# Expectations of reward and efficacy guide cognitive control allocation

R. Frömer [1,4 ✉], H. Lin [2,4 ✉], C. K. Dean Wolf[1], M. Inzlicht[2,3] & A. Shenhav[1]

The amount of mental effort we invest in a task is influenced by the reward we can expect if we perform that task well. However, some of the rewards that have the greatest potential for driving these efforts are partly determined by factors beyond one's control. In such cases, effort has more limited efficacy for obtaining rewards. According to the Expected Value of Control theory, people integrate information about the expected reward and efficacy of task performance to determine the expected value of control, and then adjust their control allocation (i.e., mental effort) accordingly. Here we test this theory's key behavioral and neural predictions. We show that participants invest more cognitive control when this control is more rewarding and more efficacious, and that these incentive components separately modulate EEG signatures of incentive evaluation and proactive control allocation. Our findings support the prediction that people combine expectations of reward and efficacy to determine how much effort to invest.

[1] Cognitive, Linguistic, and Psychological Sciences, Carney Institute for Brain Science, Brown University, Providence, RI, USA. [2] Department of Psychology, University of Toronto, Toronto, Canada. [3] Rotman School of Management, University of Toronto, Toronto, Canada. [4] These authors contributed equally: R. Frömer, H. Lin. ✉email: romy_fromer@brown.edu; hauselin@gmail.com

Cognitive control is critical to one's ability to achieve most goals[1-3]—whether to complete a paper in time for its deadline or to send that birthday message amidst a busy workday—but exerting control appears to come at a cost. We experience cognitive control as mentally effortful[4,5], and therefore require some form of incentive to justify investing control in a task[6,7]. For instance, a student is likely to study harder for an exam that has higher stakes (e.g., worth half of their grade) than a lower-stakes exam. Accordingly, research has shown that participants generally exert more mental effort on a cognitive control task (e.g., Stroop, flanker) when they are offered higher rewards for performing well, as evidenced by improved task performance and increased engagement of relevant control circuits[7-16].

In the real world, increased control may not always translate to achieving desired outcomes. For instance, even when the stakes are high, that same student is likely to exert less effort studying if they think that those efforts have little bearing on their score on that exam (i.e., that their efficacy is low), say if they felt that grading for the exam is driven mostly by factors out of their control (e.g., subjectivity in grading, favoritism). While previous work has closely examined the mechanisms by which people evaluate the potential rewards to expect for a certain control allocation, much less is known about how they evaluate the efficacy of that control, nor how these two incentive components (reward and efficacy) are integrated to determine how much control is invested.

We have recently developed a model that formalizes the roles of evaluation, decision-making, and motivation in allocating and adjusting cognitive control[17,18] (Fig. 1). Our model describes how cognitive control can be allocated based on the overall worth of executing different types and amounts of control, which we refer to as their expected value of control (EVC). The EVC of a given control allocation is determined by the extent to which the costs that would need to be incurred (mental effort) are outweighed by the benefits. Critically, these benefits are a function of both the expected outcomes for reaching one's goal (reward, e.g., money or praise) and the likelihood that this goal will be reached with a given investment of control (efficacy) (Fig. 1A). The amount of control invested is predicted to increase monotonically with a combination of these two incentive components (Fig. 1B).

The EVC model integrates over and formalizes past theories that posit roles for reward/utility and/or efficacy/controllability/agency in the motivation to engage in a particular course of action[19-26]. In so doing, our model enables a description of the computational and neural mechanisms of control allocation. For instance, past research has shown that behavioral and neural markers of proactive control increase with increases in anticipated task difficulty[27-32]. Through the lens of the EVC theory (Fig. 1), these difficulty-related increases in control intensity can be accounted for by changes in expected reward (i.e., the harder the task, the less likely you are to achieve the rewards associated with performing the task well) and/or changes in expected efficacy (i.e., the harder the task, the less helpful a given level of control is for achieving the same level of performance). The latter explains why the relationship between control intensity and task difficulty is nonmonotonic—once the task exceeds a certain difficulty (i.e., once the effort is no longer efficacious[33,34]), a person stops intensifying their mental efforts and instead starts to disengage from a task.

Our theory, therefore, makes the prediction that differences in efficacy (holding expected reward and difficulty constant) should itself be sufficient to drive changes in behavioral and neural signatures of control allocation. The theory makes the further prediction that reward and efficacy should shape incentive processing and associated neural correlates at multiple stages, including during the initial evaluation of each of these incentive components and at the point when those components converge to determine control allocation based on their combined value (EVC).

Here, we test these predictions across three studies using a paradigm that explicitly dissociates expectations of reward and efficacy associated with a cognitive control task (the Stroop task; Fig. 2), allowing us to isolate their individual and joint contributions to control allocation. To further examine how reward and efficacy are encoded at different stages of incentive processing, in Study 2 we measured EEG and pupillometry while participants performed this task, allowing us to separately measure the extent to which reward and efficacy are reflected in signals associated with the initial evaluation of the incentives available on a given trial (putatively indexed by the post-cue P3b[27,35]) versus those associated with the proactive allocation of the control

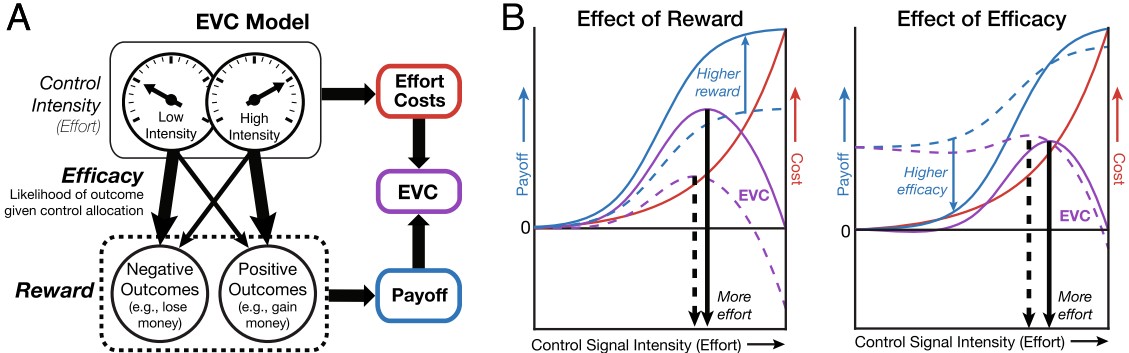

**Fig. 1 The expected value of control (EVC) model predicts that control should increase with expected reward and efficacy. A** EVC model. Control intensity is chosen to optimize the trade-off between effort costs and payoff, maximizing the expected value of the control. The payoff of a given control signal is determined by the expected reward and efficacy for a given control intensity. **B** The EVC model proposes that higher intensities of control (*x* axis) are associated with greater effort costs (red) but that these effort costs can be outweighed by the expected payoff for a given control intensity (blue). These payoffs typically increase as a function of task performance, and task performance typically improves as a function of control intensity. The EVC of each control intensity (purple) is calculated as the difference between its payoff and its cost. The optimal level of control to invest is the one that maximizes EVC (vertical black arrows). Left: When the expected reward for performing the task well is higher (from dashed to the solid line), higher control intensities achieve even higher payoffs. Right: When performance matters less for acquiring a given reward (low efficacy (dashed line))—here, simulated by having reward be unrelated to performance on most trials (instead, occurring at a fixed high rate; see Fig. 2)— the relative payoff for high vs. low control intensities decreases.

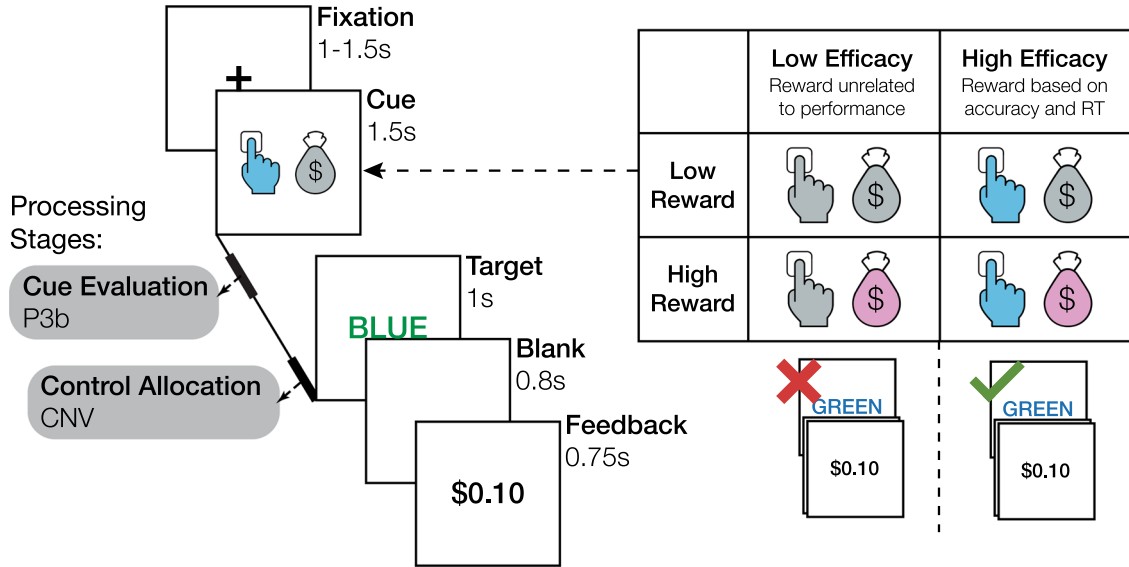

**Fig. 2 Manipulation of expected reward and efficacy.** On each trial, participants view an incentive cue followed by a Stroop stimulus (target) and then feedback indicating how much reward they received. Four different cues indicate whether a trial is high or low in reward and efficacy. The P3b and CNV event-related potentials are measured during the interval between cue and target, as indices of cue evaluation (P3b) and control allocation (CNV).

deemed appropriate for the upcoming trial (putatively indexed by the contingent negative variation (CNV) occurring prior to the presentation of the target stimulus[27,30,32,36–38]). Confirming our predictions, all three studies find that participants adaptively increase their control allocation (and thus performed better at the task) when expecting higher levels of reward and efficacy. Study 2 shows that both incentive components amplify event-related potentials (ERPs) associated with distinct stages of incentive processing: incentive evaluation (indexed by the P3b following cue presentation) and control allocation (indexed by the CNV prior to Stroop target onset). Critically, only the CNV reflects the integration of reward and efficacy. The amplitude of both ERPs, but more so the CNV, predicts performance when the target appears, supporting the prediction that these neural signals index different stages in the evaluation and allocation of control.

**Results**

To test the prediction that reward and efficacy together shape cognitive effort investment and task performance, we developed and validated a paradigm that manipulates efficacy independently from expected reward (Fig. 2). Specifically, prior to the onset of a Stroop stimulus (the target), we cued participants with the amount of monetary reward they would receive if successful on that trial ($0.10 vs. $1.00) and whether success would be determined by their performance (being fast and accurate; high efficacy) or whether it would instead be determined independently of their performance (based on a weighted coin flip; low efficacy). Using an adaptive yoking procedure, we held expected reward constant across efficacy levels, while also varying reward and efficacy independently of task difficulty (i.e., levels of congruency).

Participants performed this task in an experimental session that measured only behavior (Study 1; $N = 21$) or one that additionally measured EEG activity and pupillometry (Study 2; $N = 44$). Studies 1–2 had the same trial structure but differed slightly in the design of the incentive cues, the overall number of trials, and within-trial timing, and were run at different study sites (see "Methods"). Predictions for Study 2 were preregistered based on findings from Study 1 (osf.io/35akg). To demonstrate the generality of our findings beyond binary manipulations of reward and efficacy, we performed an additional behavioral study

(Study 3, $N = 35$) in which we varied reward and efficacy parametrically, across four levels of each.

**Performance improves with increasing expected reward and efficacy.** We predicted that reward and efficacy would together incentivize greater control allocation. Given that participants needed to be fast and accurate to perform well on our task, we expected to find that participants would be faster to respond correctly when they expected control to be more rewarding and more efficacious. Replicating previous findings[27], across both studies we found that reaction times on accurate trials (i.e., accurate RTs, split-half reliability based on odd vs even trials for Study 1: $r = 0.79$ and Study 2: $r = 0.91$) were faster for high compared to low reward trials (Study 1: $b = -9.81$, $P = 0.002$; Study 2: $b = -5.03$, $P = 0.004$). Critically, accurate RTs were also faster for high compared with low efficacy trials (Study 1: $b = -14.855$, $P < 0.001$; Study 2: $b = -5.89$, $P = 0.016$). We further found reward-efficacy interactions in the predicted direction— with the speeding effect of reward being enhanced on high-efficacy trials—but this interaction was only significant in Study 2 (Study 1: $b = -9.75$, $P = 0.116$; Study 2: $b = -9.23$, $P = 0.009$; cf. Fig. 3). Note that Study 1 had a much smaller sample size than Study 2 and 3, and therefore may not have been sufficiently powered to secure the interaction effect.

Additional analyses confirmed that these performance improvements were not driven by speed-accuracy tradeoffs. Whereas participants were faster when reward or efficacy was high, they were not less accurate (Supplementary Tables 1 and 2). In fact, their accuracies (split-half reliability Study 1: $r = 0.72$, Study 2: $r = 0.83$) tended to improve when reward or efficacy was high, though only the effect of efficacy on accuracy in Study 2 was significant ($b = 0.08$, $P = 0.033$, Supplementary Table 2). Together, the faster RTs and more accurate responses suggest that the effects of reward and efficacy on response speed reflected increased control rather than a lowering of response thresholds (i.e., increased impulsivity).

All of the analyses above control for the effects of task difficulty (response congruence) and practice effects (trial number) on performance, which in both studies manifested as worse performance (slower and less accurate responding) with increasing response incongruence, and improved performance (faster

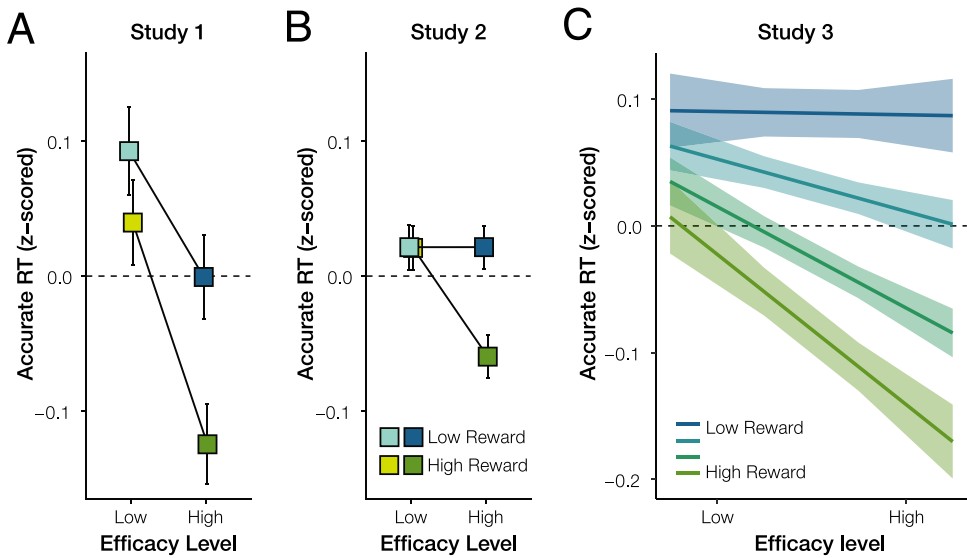

**Fig. 3 Reward and efficacy improve performance.** Across Studies 1–3, participants were fastest to give a correct response when both reward and efficacy were high, suggesting that these variables jointly determined control allocation. Panels **A** and **B** show average RTs for each of the four conditions when the reward ($0.10 or $1.00) and efficacy (0% or 100%) were varied dichotomously in Study 1, $n = 21$ participants (**A**) and Study 2, $n = 44$ participants (**B**) (see Fig. 2). Error bars represent the within-subject standard error of the mean. Panel **C** shows the estimated effect of parametrically varied levels of reward ($0.10, $0.20, $0.40, or $0.80) and efficacy (25%, 50%, 75%, or 100%) in Study 3, $n = 35$ participants, based on a linear mixed-effect model. For visual comparison with panels **A** and **B**, this is shown based on a fit to within-subject z-scored accurate RT (therefore omitting random intercepts), but the statistics reported in the text are based on the full model. Shaded error bars represent the standard error of the mean. Source data are provided as a Source Data file.

**Table 1 Effects of reward and efficacy on neural signals.**

| Predictors | P3b | | | CNV | | |
|---|---|---|---|---|---|---|
| | **Estimates** | **CI** | **P** | **Estimates** | **CI** | **P** |
| Intercept | 0.26 | −0.27 to 0.78 | 0.339 | −0.16 | −0.63 to 0.31 | 0.507 |
| Efficacy | 0.44 | 0.24 to 0.65 | **<0.001** | −0.30 | −0.53 to −0.08 | **0.008** |
| Reward | 0.34 | 0.16 to 0.51 | **<0.001** | −0.28 | −0.45 to −0.11 | **0.001** |
| Trial | −0.54 | −0.62 to −0.46 | **<0.001** | −0.03 | −0.12 to 0.05 | 0.461 |
| P3b baseline | 0.00 | −0.01 to 0.01 | 0.950 | | | |
| Efficacy:reward | −0.01 | −0.33 to 0.31 | 0.948 | −0.35 | −0.69 to −0.01 | **0.046** |
| P3b | | | | 0.16 | 0.14 to 0.17 | **<0.001** |
| CNV baseline | | | | −0.02 | −0.03 to −0.00 | **0.005** |
| Observations | 22,580 | 22,580 | | | | |

Statistics are derived from linear mixed-effects models with predictors as noted. Statistically significant P values (<0.05, two-sided) are shown in bold.

and more accurate responding) over time (Supplementary Tables 1 and 2 and Supplementary Fig. 1). The effects of reward and efficacy on performance did not significantly differ between the two studies (Supplementary Table 3).

We further replicated and extended these findings in Study 3, in which reward and efficacy were varied parametrically rather than only across two levels each. As in Studies 1 and 2, we found that participants were faster to respond correctly with increasing expected reward ($b = -7.02$, $P < 0.001$) and increasing expected efficacy ($b = -3.85$, $P = 0.001$), and that these two incentive components interacted ($b = -2.27$, $P = 0.027$), such that participants responded fastest when both reward and efficacy were highest (Fig. 3C). As in the previous studies, these effects were not explained by speed-accuracy tradeoffs. In all analyses, we controlled for task difficulty and practice effects (Supplementary Table 4).

**Reward and efficacy levels are reflected in neural signatures of cue evaluation and control allocation.** Our behavioral results

suggest that participants adjust their mental effort investment (allocation of cognitive control) based on the expected rewards and the degree to which this effort is perceived to be efficacious; they invest more effort when the expected reward and efficacy are high. To examine the neural and temporal dynamics associated with the processing of these two incentive components, we focused on two well-characterized event-related potentials (ERPs): the P3b (split-half reliability $r = 0.86$), which peaks around 250–550 ms following cue onset and is typically associated with cue evaluation[27], and the CNV (split-half reliability $r = 0.75$), which emerges about 500 ms prior to Stroop target onset and is typically associated with preparatory attention or proactive control[27,28,30,32,36,37]. Based on past research, we preregistered the predictions below for the CNV. Additional predictions regarding the P3b were generated subsequent to preregistration based on further review of the literature.

We found that reward and efficacy modulated both of these ERPs (Table 1 and Fig. 4). Replicating past work[27], cues signaling higher rewards were associated with significantly larger amplitudes of both P3b ($b = 0.34$, $P < 0.001$) and CNV ($b = -0.28$,

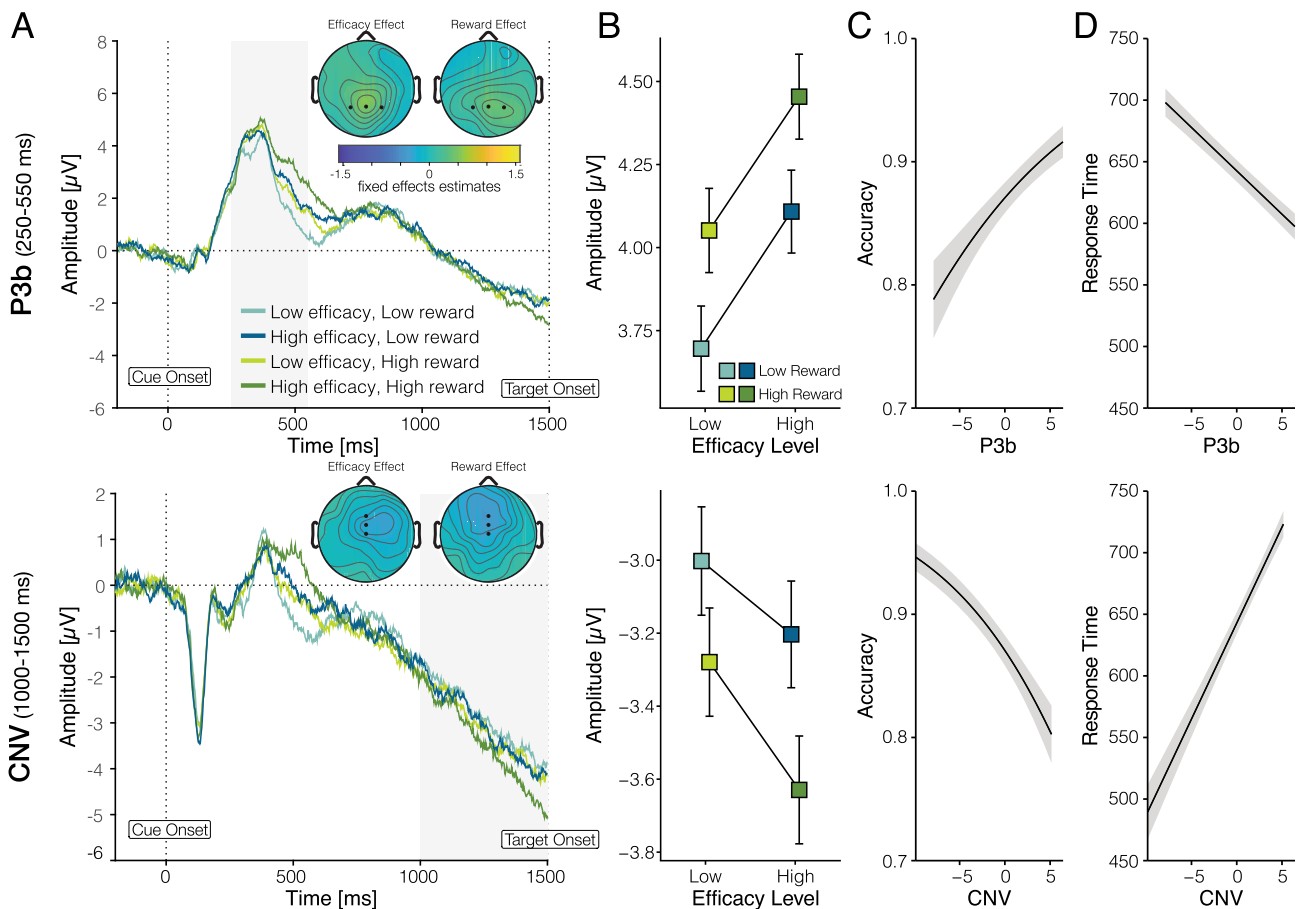

**Fig. 4 P3b and contingent negative variation (CNV) amplitudes track increases in expected reward and efficacy. A** ERP averages for each incentive condition, separately for the P3b (top) and CNV (bottom). Shaded areas indicate time windows used for quantification. Topographies show group-level (fixed-effect) contrasts for reward and efficacy, respectively. **B** Average ERP amplitudes within the relevant time windows show that the magnitude of the (positive-going) P3b and the (negative-going) CNV increase with greater reward and efficacy. The CNV but not the P3b tracked the interaction of reward and efficacy, mirroring the behavioral effects of these incentives (cf. Fig. 3). Error bars represent the within-subject standard error of the mean. **C**, **D** Fixed (group-level) effects of ERP amplitude on accuracy (**C**) and accurate RT (**D**) show that increased P3b and CNV predicted better performance when the target appeared. Shaded error bars represent the standard error of the mean. **A**–**D** $n = 44$ participants. Source data are provided as a Source Data file.

$P = 0.001$). Importantly, holding reward constant, cues signaling high rather than low efficacy were likewise associated with significantly larger amplitudes of P3b ($b = 0.44$, $P < 0.001$) and CNV ($b = -0.30$, $P = 0.008$).

Crucially, only the CNV tracked the interaction of reward and efficacy ($b = -0.35$, $P = 0.046$), with the effect of reward on CNV amplitude being enhanced when efficacy was high. We did not find a significant reward-efficacy interaction for the P3b ($b = -0.01$, $P = 0.947$). Thus, although reward and efficacy independently modulated the P3b and CNV (i.e., main effects of reward and efficacy on both ERPs), only the CNV reflected their integration (i.e., reward-efficacy interaction). This pattern of results is consistent with our prediction that reward and efficacy are initially evaluated separately (reflected in the P3b), but are subsequently integrated to determine EVC and thereby allocate control (reflected in the CNV).

**Neural signatures of incentive processing predict effort investment.** We have shown that reward and efficacy affect behavioral performance (accurate RT) and neural activity during initial cue evaluation (P3b) and proactive control allocation (CNV), suggesting that these neural signals reflect the transformation of incentives into effort allocation. To test this hypothesis more directly, we included single-trial P3b and CNV amplitude

(normalized within-subject) as regressors in our models of accurate RT and accuracy, to test whether variability in these two neural signals explained trial-by-trial variability in task performance (Table 2). We found that both P3b and CNV were associated with better Stroop task performance when the target appeared: larger ERP magnitudes were associated with an increased probability of responding correctly (P3b; $b = 0.08$, $P < 0.001$, CNV: $b = -0.10$, $P < 0.001$), and also with faster accurate RTs (P3b: $b = -7.04$, $P < 0.001$, CNV: $b = 15.58$, $P < 0.001$). Crucially, the CNV's relationship with accurate RT was significantly stronger than the P3b's ($X^2 = 18.51$, $P < 0.001$), providing evidence consistent with our prediction that the CNV plays a more important role in allocating control than the P3b, and with our observation that CNV's relationship with reward and efficacy more closely resembles that found for accurate RT (i.e., both the CNV and accurate RT were modulated by the interaction of reward and efficacy; compare Fig. 3 and Fig. 4B). However, CNV and P3b did not differ reliably in their association with accuracy ($X^2 = 0.78$, $P = 0.378$). Both ERPs further explained variance in behavior when examining each incentive condition separately (Supplementary Table 5), suggesting that these neural markers did not merely covary with behavior through shared variance with incentives. Together, these findings suggest that the P3b and CNV index the transformation of incentive processing into effort investment, a process that entails

**Table 2 Effects of neural signals on behavioral performance.**

| Predictors | Accuracy | | | Accurate RT | | |
|---|---|---|---|---|---|---|
| | Log odds | CI | P | Estimates | CI | P |
| Intercept | 1.87 | 1.65 to 2.08 | **<0.001** | 647.66 | 630.60 to 664.71 | **<0.001** |
| Efficacy | 0.08 | 0.00 to 0.15 | **0.044** | −4.00 | −8.84 to 0.85 | 0.106 |
| Reward | 0.03 | −0.05 to 0.10 | 0.465 | −3.25 | −6.85 to 0.34 | 0.076 |
| P3b | 0.08 | 0.04 to 0.11 | **<0.001** | −7.04 | −8.87 to −5.21 | **<0.001** |
| CNV | −0.10 | −0.14 to −0.06 | **<0.001** | 15.58 | 12.02 to 19.15 | **<0.001** |
| Congruency (i–n) | 0.65 | 0.44 to 0.85 | **<0.001** | −64.00 | −68.85 to −59.14 | **<0.001** |
| Congruency (n–c) | 0.34 | 0.18 to 0.49 | **<0.001** | −15.86 | −20.27 to −11.45 | **<0.001** |
| Baseline | 0.02 | −0.02 to 0.07 | 0.326 | 2.23 | 0.05 to 4.42 | **0.045** |
| Trial | 0.04 | 0.00 to 0.08 | **0.036** | −10.77 | −12.58 to −8.95 | **<0.001** |
| Efficacy:reward | 0.01 | −0.14 to 0.17 | 0.853 | −9.96 | −17.14 to −2.78 | **0.007** |
| Observations | 22,580 | | | 18,999 | | |

Statistics are derived from linear mixed-effects models with predictors as noted. Statistically significant P values (<0.05, two-sided) are shown in bold. Congruency (n–i) refers to the comparison between incongruent and neutral Stroop stimulus; Congruency (c–n) refers to the comparison between neutral and congruent Stroop stimulus.

the integration of the reward and efficacy expected on a given trial.

**Opposing effects of expected efficacy on pupil responses**. Contrary to our predictions, we observed no effect of reward on pupillary responses to the cue, and pupil responses were smaller (not larger) when efficacy was higher ($P < 0.001$, cf. Supplementary Table 6, Supplementary Fig. 2). We did not find a significant interaction between reward and efficacy on pupil diameter. These patterns of pupil responses therefore diverge from the patterns we observed in performance and ERP magnitudes (both of which scaled positively with reward and efficacy), but they nevertheless provide evidence against the idea that our behavioral and EEG results merely reflect changes in arousal. If this alternative explanation were true, pupil diameter—a reliable index of arousal[39–41], previously shown to scale with uncertainty[42]—should have increased when either reward or efficacy was high.

**Influences of incentives on EEG signatures of response and feedback monitoring**. While the focus of our study was on measures of incentive processing and proactive control allocation, we preregistered secondary hypotheses regarding the potential influence reward and efficacy might have on neural signatures of reactive control. Specifically, we predicted that these incentive components might enhance monitoring of response accuracy and subsequent feedback. Contrary to this hypothesis, when examining the error-related negativity (ERN)—a negative deflection in response-locked activity for errors relative to correct responses[43,44] (though see ref. [45] for ERN elicited by partial errors on correct trials)—we did not find main effects of reward or efficacy ($Ps > 0.444$) but did find a significant interaction ($b = 1.52$, $P = 0.001$; Supplementary Fig. 3; Supplementary Tables 7–9), whereby ERN amplitude on error trials was greatest (i.e., most negative) on trials with low reward and low efficacy (see also Supplementary Table 10 for complementary analyses of mid-frontal theta). Follow-up analyses suggest that this pattern may result from different dynamics in control and response evaluation between conditions (see Supplementary Fig. 3 and Supplementary Tables 7–9).

We found a different pattern of results when examining the feedback-related negativity (FRN), which typically indexes the difference in feedback-locked activity for trials that resulted in negative compared to positive feedback[46]. Consistent with previous findings[47–52], we found a reliable effect of receipt vs omission of reward on FRN amplitude ($b = 0.80$, $P < 0.001$), and this effect was enhanced for high reward trials ($b = 0.81$,

$P = 0.007$; Supplementary Table 11). However, in addition to this, and contrary to the hypothesis we preregistered based on previous findings[53,54], we found that effects of reward receipt vs omission on FRN amplitude were reduced for trials with high efficacy compared to those with low efficacy ($b = −0.83$, $P = 0.007$; Supplementary Fig. 4 and Supplementary Table 11). As we elaborate on in the Supplementary Discussion, this efficacy-related FRN finding might reflect the fact that, under conditions of low efficacy, reward outcomes are less predictable, thus weakening predictions about forthcoming reward.

## Discussion

Cognitive control is critical but also costly. People must therefore choose what type and how much control is worth exerting at a given time. Whereas previous studies have highlighted the critical role of expected rewards when making these decisions, our studies highlight a further determinant of control value that is equally critical: how efficacious one perceives their efforts to be (i.e., how much does intensifying control increase their chances of obtaining their reward). Across two studies, we showed that participants were sensitive to both expected reward and efficacy when determining how much control to allocate, and therefore performed best when expecting higher levels of reward and efficacy. Study 2 further demonstrated that both incentive components increase distinct ERPs, separately related to cue evaluation and proactive control, providing markers of different stages of incentive processing (evaluation vs. allocation of control). Collectively, these findings lend support to our theory that participants integrate information relevant to determining the EVC, and adjust their control allocation accordingly.

Previous research has shown that people often expend more effort on a task when it promises greater reward[7–16], but this effort expenditure has its limits. If obtaining that reward also requires greater effort (i.e., if the higher reward is also associated with greater difficulty), the individual may decide not to invest the effort[55]. Similarly, if difficulty remains constant, but reward becomes less contingent on effort (i.e., efficacy decreases), the individual may again decide to divest their efforts[33,34,56,57]. The EVC theory can account for all of these phenomena, and predicts that expected reward and efficacy will jointly determine how mental effort is allocated (in the form of cognitive control) and that the underlying evaluation process will unfold from initial cue evaluation to eventual control allocation. Specifically, the theory predicts that these incentive components will be processed sequentially over multiple stages that include initial evaluation of each component, their integration, control allocation, and

execution of the allocated control. Our behavioral and neural findings validate the predictions of this theory: When participants expected their control to have greater reward and efficacy, we saw increased neural activity in consecutive ERPs associated with incentive evaluation (P3b) and control allocation (CNV), followed by increases in control (reflected in improved performance).

Our EEG results extend and clarify previous findings. First, in a previous study, the cue-locked P3 tracked the expected reward, but not difficulty, on that cued trial[27]. We varied expected efficacy while holding expected difficulty constant, and show that varying efficacy alone is sufficient to generate comparable increases in the cue-locked P3b as variability in expected reward. The difference between our finding and the null result previously observed for task difficulty may be accounted for by the fact that efficacy (like reward) has a monotonic relationship with motivational salience, whereas difficulty does not (as discussed further below).

Second, our results extend previous studies that linked the CNV with preparatory attention and proactive control[58–60]. CNV amplitudes scale with a cue's informativeness about an upcoming task[28,36,37,61,62], temporal expectations about an upcoming target[63–65], and an individual's predicted confidence in succeeding when the target appears[66]. Critically, CNV amplitudes also scale with the expected reward for and difficulty of successfully performing an upcoming task[27,30,32,38], suggesting that this component reflects adjustments to proactive control in response to motivationally relevant variables. Here, we extend this body of work by showing that the CNV not only varies with expected efficacy (when isolated from expected difficulty) but that, unlike the P3b, it is further modulated by the interaction between reward and efficacy (i.e., the expected payoff for control; Fig. 1), and predicts trial-to-trial variability in performance, suggesting that it may index the allocation and/or execution of control based on an evaluation of EVC.

With that said, we note that variability in performance was also associated with P3b amplitude, though to a somewhat lesser degree. While it is, therefore, possible that control allocation was already being determined at this early stage of the trial, past findings[67] as well as our current ones suggest that it is equally or perhaps more likely that the P3b indexed the initial evaluation of the motivational relevance of cued incentives, as we originally hypothesized. Consistent with our original interpretation, we found that the amplitude of the P3b (but not CNV) decreased over the course of the session, potentially reflecting decreased attentiveness to the cues. It is further of note that even both ERP components combined did not fully mediate incentive effects on performance. This could be due to those ERPs being noisy indicators of the underlying processes, or to dynamics following target onset that lead to additional variance as a function of incentives that cannot be explained with proactive control. Specific predictions of the latter account could be tested in future work explicitly designed to do so.

Our remaining findings provide evidence against alternative interpretations of these neural results, for instance, that they reflect increased arousal or overall engagement throughout the trial. Pupil diameter, an index of arousal[40–42], was larger when participants were expecting lower efficacy. Although this pattern was not predicted in advance, these findings are consistent with the interpretation that pupil responses in our paradigm track arousal—induced by higher uncertainty under low efficacy—instead of proactive control[39,42]. In contrast, the magnitudes of the P3b and CNV increased with both reward and efficacy, suggesting that these two ERPs reflect processes related to proactive control rather than changes in arousal.

Our response- and feedback-related results further suggest that reward and efficacy specifically increased proactive control, but not reactive control (performance monitoring[68]) or overall engagement. Unlike the P3b and CNV, indices of performance monitoring (the ERN and FRN) were not enhanced with greater reward and efficacy, suggesting that these incentive conditions were not simply amplifying the motivational salience of errors and reward outcomes. Rather than reflecting motivational influences on control, the unique patterns of ERN and FRN amplitudes we observed across conditions may instead provide insight into how participants formed and updated expectations of performance and outcomes across these different conditions[69] (see Supplementary Discussion).

Our study builds on past approaches to studying interactions between motivation and cognitive control[9,12,16,38] by examining changes in effort allocation in response to two incentive components that are predicted to jointly increase one's motivation. Thus, unlike studies that only vary the expected reward for an upcoming task, our behavioral and neural findings cannot be accounted for by general increases in drive, vigor, or arousal. Further, unlike studies that vary the expected difficulty of an upcoming task, resulting in the nonmonotonic allocation of effort (the classic inverted U-shaped function of effort by difficulty[70–72]) the incentive components we varied should only engender monotonic increases in effort. The monotonic relationship between these incentive components and the value of control (EVC) can in fact account for the nonlinear effect of difficulty on effort allocation: at very high levels of difficulty, a given level of control becomes less and less efficacious. Our study, therefore, provides the most direct insight yet into the mechanisms and influences of EVC per se, rather than only some of its components.

One interesting feature of our results is that participants engaged in some reasonably high level of effort even when their efforts were completely inefficacious (0% efficacy). There are several plausible explanations for this, including an intrinsic bias towards accuracy (or against error commission)[73] and potential switch costs associated with the interleaved trial structure[74]. For instance, switch costs associated with control adjustments may discourage a significant drop in control following a high-efficacy trial. An even more intriguing possibility is that experiences in the real-world drive participants to have strong priors that their efforts are generally efficacious (and practice allocating control within a certain range of expected efficacies)[75], making it difficult for them to adjust all the way to the expectation that reward is completely unrelated to their performance on a task.

Individual differences in expectations of efficacy may also play a significant role in determining one's motivation to engage in individual tasks or effortful behavior at large[19,21,76–78]. Forms of amotivation, like apathy and anhedonia, are common across a variety of psychiatric and neurological disorders, and most likely reflect deficits in the process of evaluating potential incentive components; determining the overall EVC of candidate control signals; specifying the EVC-maximizing control allocation; and/or executing this control. Thus, to understand what drives adaptive versus suboptimal control, we need to find new and better ways to assess what drives these key processing stages underlying motivated effort. By highlighting the crucial role efficacy plays in determining whether control is worthwhile, and identifying candidate neural signatures of the process by which this is evaluated and integrated into decisions about control allocation, our studies pave the way toward this goal.

## Methods
### Study 1

*Participants*. In total, 21 individuals participated in Study 1 (age: M = 21.14, SD = 5.15; 17 female). Participants gave informed consent and received partial course credits and cash ($5 to $10, depending on their performance and task

contingencies) for participation. The study was approved by Brown University's Institutional Review Board.

*Design and procedure.* We used a within-subject 2 (reward: high, low) × 2 (efficacy: high, low) design. On high and low reward Stroop trials, participants saw cues that informed them that they would receive $1.00 and $0.10, respectively, on the upcoming trial (Fig. 2). Reward levels were crossed with efficacy. On high-efficacy trials, whether participants were rewarded depended entirely on their performance (i.e., fast and accurate responses were always rewarded—100% performance–reward contingency, cf. Supplementary Table 12 for summary statistics on criterion-performance and reward). On low efficacy trials, rewards were not contingent on participants' performance; instead, rewards were sampled from a rolling window (size = 10) of reward rate in high-efficacy trials to match reward rates across efficacy levels. This approach parallels and builds on recent work examining the influence of performance contingency in the domain of motor control (where individuals simply needed to respond quickly[56,57], see also refs. [79,80]), but importantly our task required that participants engage cognitive control in order to be fast while also overcoming a prepotent bias to respond based on the color word.

Participants first completed three practice blocks. In the first practice block (80 trials), participants learned the key-color mappings by indicating whether the stimulus XXXXX was displayed in red, yellow, green, or blue (using D, F, J, K keys; counterbalanced across participants). In the second practice block (16 trials), participants learned to associate cues with different reward and efficacy levels (Fig. 2). Finally, participants completed a practice block (64 trials) that resembled the actual task. Incentive instructions read as follows: "In the next block, you again need to press the key associated with the color of the text on the screen. From now on, you will have the opportunity to get an additional bonus based on how you perform the task. You will be told on each trial how performance could affect your bonus. Before each word appears, you will see an image that tells you two things: (1) the amount of reward you could earn; and (2) whether or not your performance will determine if you get that reward. When you see one of the two images above, you can get a low ($0.10) or high reward ($1.00) if you respond quickly and accurately. The two images above ALSO indicate that you can get a low or high reward, BUT the gray hands indicate that your reward will have NOTHING to do with how quickly or accurately you perform. Instead, these rewards will be determined randomly. As long as you provide some response on that trial, you have some possibility of getting a low ($0.10) or high ($1.00) reward. Although these rewards will be random, you will be just as likely to get a reward on these trials as the trials with the blue hands."

Once familiar with the task, participants were introduced to the performance bonus and completed the main task. Performance bonus instructions read as follows: "From now on, you will continue performing the same task, but it will not be practice. Every trial can influence your ultimate bonus. At the end of the session, we will choose ten trials at random and pay you an additional bonus based on the total amount of money you earned across those ten trials. This means you have the opportunity to earn up to ten additional dollars on this task." On an individual trial, cues were presented for 1500 ms, followed by a 250 ms fixation cross, followed by a target. To increase task difficulty, the response deadline for each trial was 750 ms but reaction times were recorded as long as a response was made within 1000 ms after Stroop stimulus onset. Immediately after a response was made, the feedback was presented for 750 ms. If a response was made before 1000 ms, the remaining time was added to the inter-trial-interval, in which a fixation cross was displayed for 500 ms to 1000 ms. The main task consisted of four blocks of 75 trials each (except for the first 14 participants, who completed 80 trials per block). The experiment was run using custom code in Matlab and the Psychophysics toolbox.

After completing the task, participants completed questionnaires that were administered for analyses unrelated to the present studies. At the end of the experiment, ten trials were randomly chosen and participants received a bonus that was the summed outcomes of those trials.

## Study 2

*Participants.* Before data collection, we conducted a sensitivity analysis, which indicated that a sample size of $N = 50$ will provide 80% statistical power to detect effect sizes of $d = 0.3$ or larger. We preregistered our sample size, task design, and analysis plan (osf.io/35akg) and recruited 53 undergraduate students (age M = 20.18, SD = 2.30; 15 male; 38 female). We excluded from all analyses 9 participants who performed poorly on the Stroop task (i.e., below 60% accuracy on high-efficacy trials), leaving 44 participants in the final sample. Technical issues also prevented us from recording clean pupil data from 7 participants in this final sample, leaving 37 participants in the pupil analyses. Participants gave informed consent and received partial course credits and cash ($5 to $10, depending on their performance and task contingencies) for participation. The study was approved by Brown University's Institutional Review Board.

*Design and procedure.* The behavioral paradigm and procedures were similar to those in Study 1. In addition, we recorded EEG and pupillary responses and changed the following task parameters: no fixation cross was presented during the cue-target interval; that is, the cue transitioned directly to the target to avoid inducing visual evoked potentials that would influence the amplitude of the CNV;

we added a post-response blank screen (800 ms) to dissociate response evaluation and feedback processing; participants performed eight blocks of 75 trials each. We also changed the appearance of the cues as depicted in Fig. 2. We selected putatively equiluminant colors (gray: C:30.98, M: 19.61, Y: 20.78, K: 0; pink: C: 9.8, M: 42.75, Y: 0, K: 0, blue: C: 61.96, M: 0, Y: 0.39, K: 0). Luminance (computed post-hoc for the four cue stimuli as a whole) was similar across the individual cue stimuli (low efficacy low reward 1.4199 cd/m², low efficacy high reward: 1.3980 cd/m², high-efficacy low reward: 1.3829 cd/m², high-efficacy high reward: 1.3577 cd/m²) and approximately 1.4 cd/m². We used the same stimuli throughout and did not counterbalance. Note that the small deviations in luminance do not correspond to the observed patterns in pupil dilation.

*EEG recording and preprocessing.* EEG data were recorded from 32 Ag/AgCl electrodes embedded in a stretched Lycra cap (Electro-Cap International, Eaton, OH) at a sampling rate of 512 Hz. Impedances were kept below 5 kΩ during recording. Vertical electrooculography (VEOG) was recorded from two electrodes placed above and below the right eye, respectively. Signals were amplified using ANT TMSi Refa8 device (Advanced Neuro Technology, Enschede, The Netherlands), grounded to the forehead, and referenced online to the average of all electrodes. Offline, the EEG data were re-referenced to the average of electrodes placed on the two earlobes. During preprocessing, continuous data were high-pass filtered at 0.1 Hz (12 dB/oct, zero phase-shift Butterworth filter) and decomposed into independent components using the infomax independent component analysis algorithm implemented in EEGLAB[81]. We inspected the independent components and used the ICLabel EEGLAB extension[82] to help identify and remove blink and noise components. We used ICLabel, an extension made by EEGLAB's developers[82], to identify ICs that were classified as eye or muscle ICs. The algorithm assigns probabilities to seven categories: brain, muscle, eye, heart, line noise, channel noise, other. The extension also provides an interface (see https://sccn.ucsd.edu/wiki/ICLabel) that shows the topography, time course, power spectrum, and ERP-image (sorted by trial number) of each IC. Guided by ICLabel's classification algorithm, for each participant, we excluded, on average, two to three eye frontal components (e.g., blinks, vertical/horizontal eye movements) and one to three muscle components (usually ICs that showed maximal activity at temporal sites). ICs were considered blinks or eye movement ICs and excluded if (1) there was a high probability (>85% and <1% brain) of them being classified as an eye-related IC and (2) the IC time course activity resembled blinks or vertical/horizontal eye movements (i.e., activity that looks like step-functions) and (3) the topography showed maximal activity at frontal sites (see https://sccn.ucsd.edu/wiki/ICLabel for an example of such an IC). ICs were considered as muscle ICs and excluded if (1) there was a high probability (>95% muscle and <1% brain) of them being classified as a muscle IC and (2) the power spectrum resembled noise or muscle activity more than neural activity (i.e., power peaks at higher frequencies rather than lower frequencies).

Pre-processed EEG data were epoched relative to the onset of four events: cue (−200 to 1500 ms), stimulus (−200 to 800 ms), response (−200 to 800 ms), and feedback (−200 to 800 ms). All epochs were baseline-corrected using the mean amplitude before event onset (−200 to 0 ms), and single-trial baseline activity was included as covariates in the statistical models[83]. Epochs containing artifacts, with amplitudes exceeding ± 150 μV or gradients larger than 50 μV, were excluded from further analysis. We focused our analyses on these event-related potentials, quantified agnostic of condition with ROIs and time windows determined a priori based on the literature[84] cue-locked P3b (250–550 ms, averaged across Pz, P3, and P4[27,85]), cue-locked late CNV (1000–1500 ms post-cue, i.e., −500 to 0 ms pre-target, averaged across Fz, FCz, and Cz[30]), response-locked correct- and error-related negativities (CRN/ERN; 0–100 ms[43,86]), and feedback-locked FRN (quantified peak-to-peak at FCz as the difference between the negative peak between 250 and 350 ms and the positive peak in the preceding 100 ms from the detected peak[47]). All EEG data preprocessing were performed using custom MATLAB scripts using EEGLAB functions (cf. [87]). For each ERP (except the FRN that was quantified peak-to-peak), we averaged the amplitudes within the specified time window separately for each epoch and exported these single-trial values for further analyses in R.

*Pupil recording and preprocessing.* Pupil data were recorded using the EyeLink 1000 Desktop Mount eye tracker (SR Research, Mississauga, Ontario, CA). The EyeLink system was configured using a 35-mm lens, 5-point gaze location calibration, monocular right-eye sampling at 500 Hz, and centroid fitting for pupil area recordings. All data processing was performed using custom R and Python scripts. Blink artifacts detected using the EyeLink blink detection algorithm were removed and subsequently interpolated linearly from −200 ms prior to and post-blink onset. Finally, we down-sampled the continuous data to 20 Hz and z-score normalized (within-subject) each data point by subtracting the mean pupil size of all data points and then dividing by the standard deviation.

## Study 3

*Participants.* In total, 35 individuals participated in Study 3 (age: M = 20.66, SD = 2.61; 27 female). Participants gave informed consent and received partial course credits and cash ($5–$10, depending on their performance and task contingencies)

for participation. The study was approved by Brown University's Institutional Review Board.

*Design and procedure.* The overall procedure was the same as in Study 1, except that expected reward and efficacy were varied parametrically across 4 levels each. As in Studies 1–2, reward levels were varied in terms of the monetary outcome at stake: $0.10, $0.20, $0.40, or $0.80. Efficacy was varied in terms of the likelihood of the outcome being determined by performance (i.e., by meeting the speed and accuracy criterion) versus being determined at random (cf. [88]), with 100% efficacy being identical to the high-efficacy condition in Studies 1–2. The possible efficacy levels were 25%, 50%, 75%, and 100%. Reward and efficacy levels were varied independently across 300 total trials. The expected reward and efficacy levels for the upcoming trial were cued by two charge bars that were filled to the current level of each.

**Analysis.** Classical frequentist statistical analyses were performed in R. The design was entirely within-subjects; unless stated otherwise, all estimates and statistics were obtained by fitting mixed-effects (multilevel or hierarchical) single-trial regression models (two levels: all factors and neurophysiological responses for each condition were nested within participants) with random intercepts and slopes (unstructured covariance matrix) using the R package lme4[89]. Random effects were modeled as supported by the data, determined using singular value decomposition, to avoid overparameterization and model degeneration[90,91]. All analysis code for reproducing the reported results can be found on OSF: osf.io/xuwn9. All continuous regressors were within-subject mean-centered. Two-tailed probability values and degrees of freedom associated with each statistic was determined using the Satterthwaite approximation implemented in lmerTest[92]. We inspected Q–Q plots for violations of normally distributed residuals and assured that there was no problematic collinearity between regressors. Wherever relevant, we also reported split-half reliabilities (correlations) based on odd/even-numbered trials.

*Behavioral.* In both samples, accuracy was analyzed using generalized linear mixed-effect models with a binomial link function. The predictors or regressors were reward, efficacy, their interaction, congruency, and trial number. Accurate RTs were modeled using linear mixed-effects models and the same predictors, and z-scored within subject for visualization, only. For trial-by-trial predictions of performance with ERPs, the behavioral models were extended by including P3b and CNV amplitudes z-scored within participants as predictors. In separate analyses, we confirmed that similar results are obtained using a step-wise approach, analyzing residuals of the behavioral model with CNV and P3b as predictors or analyzing incentive effects on residuals from a model with ERPs but without incentive conditions, suggesting partially non-overlapping variance. Trial number (Trial) was added to these and all other models as a nuisance regressor to account for trends over time, such as learning or fatigue effects (cf. Supplementary Fig. 5).

*EEG.* Full linear mixed-effect models for all ERPs included reward, efficacy, and their interactions, as well as trial as predictors. For each ERP, we regressed out the baseline activity at the same electrode sites[83]. This approach accounts for variability prior to the effect of interest that can otherwise induce spurious effects due to noise or spill-over from previous stages of the trial. Although noise in the baseline is assumed to average to zero (across time points, as well as trials) when using traditional ERP-averaging approaches, this assumption does not necessarily hold for single-trial analyses, where a non-stationary baseline or unevenly distributed noise can easily lead to systematic biases in the subsequent time-series. To address these potential spurious effects, we follow recommendations to include the baseline as a nuisance regressor[83]. In the CNV analyses, we further controlled for variation in the preceding P3b amplitude, because here, likewise, due to the autocorrelation of the signals, larger P3b amplitudes (a positive-going ERP) would require larger subsequent CNV amplitudes (a negative-going ERP) to counteract the larger positive P3b amplitudes and reach the average levels of CNV amplitude. We compared the results with and without the inclusion of the P3b as a regressor and the patterns of results were qualitatively similar. For the ERN analyses, we included as predictors target congruency, response accuracy, and interactions with incentives. For FRN analyses, we included as predictors the outcome (whether trials were rewarded or not), and interactions with incentives.

*Pupil response model and analysis.* We modeled the pupillary response as a linear time-invariant system comprising of a temporal sequence of "attentional pulses"[93]. As with methods used in functional magnetic resonance imaging analysis to deconvolve blood-oxygen-level-dependent signals, this approach allows us to deconvolve temporally overlapping pupil responses and estimate the magnitude of the pupil response associated with each event. Following previous work[93,94], each event (e.g., fixation, cue, target, response, feedback) was modeled as a characteristic impulse response approximated by an Erlang gamma function, $h = t^n e^{\left(\frac{-nt}{t_{max}}\right)}$, where the impulse response $h$ is defined by $t$, the time since event onset, $t_{max}$, the latency of response maximum, and $n$, the shape parameter of the Erlang distribution. Guided by previous empirical estimates[95], we set $n = 10.1$ and $t_{max} = 1.30s$. We used the pypillometry python package to estimate the magnitude (i.e., coefficient) of the pupil response for each event[96], and we z-scored normalized these

coefficients (within-subject) before fitting mixed-effects models to evaluate whether the coefficients varied as a function of the experimental manipulations (i.e., efficacy, reward, target congruency, and feedback).

**Reporting summary**. Further information on research design is available in the Nature Research Reporting Summary linked to this article.

## Data availability
The datasets generated and analyzed during this study are available under https://osf.io/xuwn9. Source data are provided with this paper.

## Code availability
Scripts for all analyses are available through https://osf.io/xuwn9.

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

## Acknowledgements

The authors are grateful to Elizabeth Cory and Aravinth Jebanesan for assistance in data collection. This research was supported by a Center of Biomedical Research Excellence grant P20GM103645 from the National Institute of General Medical Sciences (A.S.), an Alfred P. Sloan Foundation Research Fellowship in Neuroscience (A.S.), and a grant from the Natural Sciences and Engineering Research Council of Canada (RGPIN-2019-05280) (M.I.).

## Author contributions

A.S., C.D.W., R.F., H.L., and M.I. conceived the study. C.D.W. and H.L. performed task coding. H.L. collected the data. R.F. and H.L. analyzed the data and wrote the paper. R.F., H.L., C.D.W., M.I., and A.S. edited the paper.

## Competing interests

The authors declare no competing interests.
