## [Peer Review File · Nature Communications]

Reviewer #1 (Remarks to the Author):

The present manuscript by Fromer et al. reports data from a behavioral experiment (Exp. 1), and a preregistered follow-up experiment combining EEG and pupillometry, investigating the effects of reward and efficacy in a Stroop task. The results indicate that reward and efficacy jointly determine behavior, and relying on partly dissociable neural processes related to incentive evaluation and allocation of proactive control.

The manuscript has a number of strong elements. It is generally well written, features two experiments, the second of which was preregistered (and had a substantial sample size), and the LME analysis approach (which also relates EEG activity to behavior) seems elegant. Yet, I also found a number of elements a bit problematic, a lot of the supplementary analyses not overly relevant, and was generally missing some details.

Major

1a) This is largely a comment, less than detailed criticism, but I found the implementation of efficacy a bit too brute-force. Specifically, it was implemented as all or nothing, which seems unrealistic (see also the rather artificial example that was presented on p. 3). I don't mean to say that investigating this clearest-possible difference is not relevant, but I found that it then lacks some of the possible elegance that more nuanced values might have been able to draw from the EVC model. As a side-note, I think it would be more accurate to describe the respective condition as "no efficacy" rather than "low efficacy", because it is literally zero. 1b) I appreciate the formalization of efficacy, and in particular its complete treatment in the EVC model itself, but some of the results of the present 2*2 experiment with simple extremes (high vs. no efficacy; low vs. high reward) came across as very expected (no problem per se, of course) and that a lot of data already makes these points. I don't mean to say that this renders the results uninteresting, but feel like a lot could already be inferred from earlier work. Would it be possible to juxtapose the present results with a prominent account (or data) that would have NOT predicted this outcome? If so, I feel like this might be helpful. Again, I do see this potential in the EVC, but less so in this specific test here.

2) In my view, a lot of (partly important) details were missing, which I'll try to list here:

- the colors (and luminance!) of the cues (both in general, and concerning possible counterbalancing).
- more details on how exactly participants were instructed, how the yoking procedure was described to the participants etc. (i.e., what was the participants' expectation in this regard). This might also have theoretical relevance. E.g., in experiment 1, participants behaved a bit like they didn't "get" that low efficacy meant that their performance was really entirely irrelevant. Similarly, in experiment 2, the P3b main effect of efficacy could be value-related in case participants were not sure whether reward expectation was really identical for low and high efficacy. The authors present plausible explanations for these effects, but, long story short, it would be important to know how participants were instructed in this regard and whether they believed it.
- the time-range and electrode selection for the P3b and CNV (and ERN); how exactly were those selected (avoiding circularity); or was there another approach to avoid type-1 errors that I overlooked? In particular, the (unequal) subdivision of the CNV seemed a bit odd. Unfortunately, no such selection information was preregistered (which would have resolved any possible concerns). Maybe the effects are so strong that the specific time-ranges and electrodes do not matter, but a clear description of how these elements were selected is necessary.
- some more information on the modelling approach. Specifically, I was not sure what "fixed effects" (see e.g. figure caption 3) refers to exactly, in this context. Moreover, it was not clear to me how the approach of integrating the ERP data in to the behavioral analysis accounted for the condition-level differences? I might misunderstand, but it sounded to me as if all conditions were lumped together, which could end up repackaging the same condition-level effects that were already present in both modalities. I am not sure I am clear here; I am thinking of how for fMRI data one would use parametric modulators that only account for residual variance after explaining the categorical condition differences.
- relatedly, I think it would be helpful to extend a bit on the inclusion of the single-trial baselines as covariates; since this is not standard, I would appreciate a sentence or two explaining this beyond just the provided reference. It would also be good to be a bit more explicit on what main

effects of, and interactions with "trial" (and not "trial number", if I understand correctly) then mean. Finally, depending on the motivation, why was this approach not extended to the cue-locked analysis of the pupil-size data?

- report the range of reward rates, both concerning what was paid out, and how successful participants were (i.e., how often they would have qualified for reward). Was this comparable across most participants?
- how were participants informed about the (random-trial-selection) reward scheme? I assume that they were given veritable information about the random drawing of ten trials, but how exactly was this information presented?
- a bit more information on which ICs were excluded would be helpful (how many? What did they seem to reflect?). I understand that this was guided by an algorithm, but more details on how it was used would be helpful.
- Fig. 4 could use some more image resolution; I would recommend larger maps (and labels to the hot-scale, even if this is a stats value, rather than μV), and larger sensor plots. It is really hard to see the CNV effect in particular. I would also endorse some low-pass filtering of the sensor data for cosmetic/visibility reasons.
- questionnaires are mentioned in the preregistration and the method section, but nothing beyond that (I think). I see that there were no predictions about them, so maybe they were only acquired to control for possible confounds or in the context of a larger endeavor? Please clarify (if I didn't miss it).
- please extend your discussion of the P3b (p. 16, middle paragraph) to what you think it means in your data.
- lines 495, 496: please motivate this approach more, and spell out its consequences. It sounds like it might be a good idea, but intuitively, I can also see this affect the data in problematic ways, since you assume that the P3b and CNV are inherently linked.

3) I don't think that this is huge, but I found the RT time-outs to be too narrow, not least because congruency is a factor here, and hence RT distributions might have been differentially cut off. As for the 750-ms deadline, I assume that it wasn't adjusted during the experiment (hence my question above for more details), right? If so, is it not possible that some participants found the task quite easy and others very challenging? Not a big concern given the within-subject nature of the experiment, but could have introduced an extra variance source...

4) I have significant problems with the pupil data. Beyond the surprising result pattern (as acknowledged), effects start way too early, to an extent that is implausible. Typical "psychological" pupil effects don't arise until close to 1 second (maybe 600 ms), and even the light reflex takes a bit of time. Hence, effects at 40 ms (and 220 ms) post-cue are simply implausible as event-related responses to those cues, and probably point to a problem with response overlap from the preceding trials (and hence to a non-random trial sequence). Or do I misunderstand the timing information here? I don't know whether the inclusion of the single-trial baseline activity should attenuate such problems or might exacerbate them (then again, this seems to have been applied only to the target and feedback screens), but as they stand, I don't think that the pupil data are really interpretable. I am sorry for the strong statement, and will happily revise it if the authors can point me to a point of misunderstanding, of course. [One thing the authors could try as a control analysis would be to look at target-locked congruency effects, which are well-documented in the literature as starting around 800 ms post-target]. If taken at face value, more discussion would be needed about what these results likely reflect specifically (rather than mostly what they do not reflect).

5) I had the impression that the authors were struggling with this quite a bit themselves. I have to honestly say that I am not sure what contribution of the rather elaborate supplemental material about the ERN, FRN, and fronto-central theta power can make. This relates to the likely dependence of response-locked and feedback-locked data (which has to differ as a function of efficacy, as the authors acknowledge), and, maybe even more importantly, again as identified by the authors themselves, to the fact that the cue-target set-up pretty much abolishes the possibility for classic reactive control processes. I am not sure what to suggest here, but I would consider further shortening those elements, certainly in the main manuscript.

Minor

1) The rationale for not having a fixation dot was unclear to me (lines 433/434). An offset response (of the cues) will be more problematic here than would have been the additional onset of a fixation cross, in my opinion.

Reviewer #2 (Remarks to the Author):

This manuscript by Frömer and colleagues presents results from a behavioral experiment and an EEG/pupillometry study investigating the effects of efficacy and expected reward on the investment of effort or, more generally, the recruitment of cognitive control. Using a Stroop task modified in a clever way to specifically manipulate efficacy while keeping difficulty and expected reward constant the authors show that people invest more control when this control is more rewarding and more efficacious (this is mainly reflected in RTs for correct trials). They furthermore show that both, expected reward and efficacy, modulate the P3b to the cue indicating these values and the prestimulus CNV. The CNV amplitude also reflects the interaction of both parameters. In line with the Expected Value of Control Model people appear to integrate worth and worthwhileness to determine the amount of to-be-recruited control. In addition, the authors present interesting results for performance-monitoring-related ERPs.

This is a very nice study using an elegant design. The methods are sound and state of the art and the results are convincing. The paper reads well and is easy to follow. The results advance the field of cognitive control research and are highly interesting even beyond this field. I have only a few minor comments:

1. It's just about wording/semantics: the paper seems to suggest that effort and recruitment of cognitive control have an equal meaning. Is that always the case?
2. What does the significant baseline effect for the CNV (Table 1) mean?
3. I did not fully understand the reasons for the statement in lines 337 and following that the pupil responses more closely track arousal than proactive control. More specifically, this would mean that lower efficacy (increasing pupil size) is associated with higher arousal. Why should this be the case?
4. The supplementary ERN/FRN analyses are very interesting. I just wonder whether for the analyses shown in Fig S2E there were enough trials in each subject, in particular for congruent and neutral trials. How reliable is this analysis? (it is cool though)
5. A more general question out of curiosity that may not be answered by this study and touches metacognition rather than the focus of this study: Can people perceive the difference of efficacy and difficulty, in particular, if there was no cue (or the meaning of the cue would have to be learned)?

Reviewer #3 (Remarks to the Author):

This manuscript reports the results from two experiments, the second pre-registered on the basis of the first, which were designed to test the expected value of control (EVC) hypothesis as it relates to both behavior and neurophysiological readouts as elicited by a novel task. The subject matter of this manuscript is of theoretical, and potentially also practical, significance. The methodology is sound and rigorous. The results are interesting, at times surprising, and always clearly described.

Please find my suggestions below.

Suggestions related to theoretical clarity: As written the manuscript's theoretical foundations may be somewhat opaque to the broad readership of this journal. For the below confusions, the authors

may wish to rephrase their existing language:

(1) The authors show evidence that correct responses are faster for high than low efficacy trials, which is said to be consistent with EVC, but in the supplement it is mentioned that incorrect responses are actually faster for low efficacy than high efficacy trials. The latter is interpreted as "consistent with effort minimization by, e.g., responding with any key)." If the latter interpretation were correct for low-efficacy trials, presumably one would then predict that correct responses would also be faster in low-efficacy trials. Can the authors please clarify this apparent contradiction?

(2) The authors demonstrate that reward and efficacy similarly interact for both behavior and the CNV, and moreover that (when included in the model) differences in the CNV continue to predict behavior even when accounting for the effects of reward, efficacy, and their interaction. However, it appears the converse is also true - efficacy continues to predict accuracy and correct RTs above and beyond the effect of CNV; similarly, the reward:efficacy interaction explains at least correct RTs above and beyond the effects of CNV. Can the authors please clarify if (or how) this may challenge the interpretation of CNV "indexing" the integration of efficacy and reward, as predicted by EVC?

(3) EVC would seem to make predictions about individual differences, but these are not the subject of any analysis here. In addition, the preregistration includes the collection of questionnaire data, which was presumably collected for this kind of reason. Are there systematic subject effects here that could complicate the use of a purely random effects account of subject-level variation?

Suggestions related to methodological approach:

- Given the methodological care of the manuscript it is surprising the authors do not report any measure of reliability. Can the authors please provide some estimate of reliability (e.g., split-half or odd/even)? It is acknowledged that due to the rolling window "yoking" procedure, these reliability estimates may be upwardly-biased.
- The authors appear to use a mean-centered explanatory variable for trial number to account for linear practice effects. However, it's not expected that practice effects are linear. And, the functional form of practice effects will be highly relevant for building on this work (e.g., in defining block lengths and total number of blocks). Can the authors please address this point, for example adding a supplementary graphic plotting the residual RTs (from a model that does not include trial length as an explanatory variable) so that future work in this area can be informed by the practice effects that may or may not be captured by linear effects of trial #?
- The authors allude to how they determined the random effects structure for their model (using "singular value decomposition", by which they presumably mean they added random effects only until the models could no longer be estimated). Can the authors please provide the final models they used, including random effects specifications? It would also be worth confirming to the reader that these models meet the relevant assumptions, that Q-Q plots for the models had been inspected, that there was no problematic collinearity between predictors, etc.
- Relatedly, there are many ways in which this data could trigger a variety of illuminating work, but I note that the authors plan to make the data available upon request. Since they have gone to the trouble of preregistering their work on OSF, I would strongly suggest the authors consider making their data available via the same mechanism. This is likely to increase the impact of the work significantly.

Minor points:

- Unless I missed it, the authors do not describe the procedure for creating integrated accuracy/rt z-scores. It is obvious enough how this must have been done, but it should be described explicitly in the manuscript.
- The authors do not report statistics for the interaction of efficacy and reward in the left column of Table 2 (for predictions of accuracy). Can this be included in the table or a sentence added to the note explaining why it is not reported?
- suggest rephrasing "both P3b and CNV were associated with better Stroop task performance on the upcoming trial" to "both P3b and CNV were associated with better Stroop task performance when the target appeared" or similar. I was briefly led astray in thinking that there might have been an analysis of n+1 trials in the manuscript.
- the correct citations for lme4 and lmerTest are as follows.
Kuznetsova A, Brockhoff PB, Christensen RHB (2017). "lmerTest Package: Tests in Linear Mixed Effects Models." *Journal of Statistical Software*, 82(13), 1–26. doi: 10.18637/jss.v082.i13.

Bates D, Mächler M, Bolker B, Walker S (2015). "Fitting Linear Mixed-Effects Models Using lme4." *Journal of Statistical Software*, 67(1), 1–48. doi: 10.18637/jss.v067.i01.

- At one point in the introduction the authors refer to the importance of "variability in efficacy (holding expected reward and difficulty constant)", but I believe that what is meant here is "differences in efficacy." Variability raised questions for me about the distribution of expected reward and difficulty, which is relevant to the definition of optimality (it's unclear why one may want to maximize the expected value as opposed to minimizing the lower bound; etc). Readers may better avoid this garden path if you use the simpler word "differences."

Reviewer #1 (Remarks to the Author):

The present manuscript by Fromer et al. reports data from a behavioral experiment (Exp. 1), and a preregistered follow-up experiment combining EEG and pupillometry, investigating the effects of reward and efficacy in a Stroop task. The results indicate that reward and efficacy jointly determine behavior, and relying on partly dissociable neural processes related to incentive evaluation and allocation of proactive control.

The manuscript has a number of strong elements. It is generally well written, features two experiments, the second of which was preregistered (and had a substantial sample size), and the LME analysis approach (which also relates EEG activity to behavior) seems elegant. Yet, I also found a number of elements a bit problematic, a lot of the supplementary analyses not overly relevant, and was generally missing some details.

Major

1a) This is largely a comment, less than detailed criticism, but I found the implementation of efficacy a bit too brute-force. Specifically, it was implemented as all or nothing, which seems unrealistic (see also the rather artificial example that was presented on p. 3). I don't mean to say that investigating this clearest-possible difference is not relevant, but I found that it then lacks some of the possible elegance that more nuanced values might have been able to draw from the EVC model. As a side-note, I think it would be more accurate to describe the respective condition as "no efficacy" rather than "low efficacy", because it is literally zero.

- We completely agree that focusing only on these extremes fails to capture the real-world range of expected efficacy values. We chose to use these extremes to provide a "proof of principle" since these were the first studies targeting the construct in this way. However, motivated by the goal expressed by the reviewer of providing more naturalistic comparison, we have since performed several follow-up studies in which expected efficacy is varied continuously. For instance, we recently performed a study that was similar to the current ones, but where expected reward and efficacy varied parametrically (4 levels each) rather than in a binary fashion. Preliminary results (N = 24) replicate our current ones, with performance improving parametrically with levels of both expected reward and efficacy (faster RTs on correct trials; p s < 0.002; see Table below), and a non-significant trend for a reward-efficacy interaction in the expected direction (potentially attributable to a thus-far smaller sample and the fact that these interactions are expected to be nonlinear). We are also preparing to submit a manuscript that reports findings from a follow-up EEG study (Grahek, Froemer, Shenhav, in prep) in which participants learn parametrically-varying efficacy levels over the course of the experimental session, and we replicate our current findings with respect to expected efficacy's influence on performance and CNV. We would prefer to keep the terminology as is for consistency with this and other ongoing work, but are open to changing it if the reviewer prefers. In the revised manuscript, we do now reiterate in multiple places throughout the manuscript that low efficacy is zero so this is clear to the readers.

AccRT			
Predictors	Estimates	CI	p

(Intercept)	600.73	587.37 – 614.09	<0.001
Efficacy	-4.68	-7.49 – -1.86	0.001
Reward	-8.35	-11.12 – -5.57	<0.001
Congruency n-i	-49.99	-65.20 – -34.78	<0.001
Congruency c-n	-16.71	-24.77 – -8.65	<0.001
Trial	-5.33	-8.44 – -2.21	0.001
Efficacy * Reward	-1.67	-4.19 – 0.85	0.194
Observations	5789		

1b) I appreciate the formalization of efficacy, and in particular its complete treatment in the EVC model itself, but some of the results of the present 2*2 experiment with simple extremes (high vs. no efficacy; low vs. high reward) came across as very expected (no problem per se, of course) and that a lot of data already makes these points. I don't mean to say that this renders the results uninteresting, but feel like a lot could already be inferred from earlier work. Would it be possible to juxtapose the present results with a prominent account (or data) that would have NOT predicted this outcome? If so, I feel like this might be helpful. Again, I do see this potential in the EVC, but less so in this specific test here.

- We agree with the reviewer that our behavioral findings align with expectations, but we see this as a feature rather than a bug of these studies. The goal of our work was not to prove that efficacy is relevant to motivation but to isolate and measure its influence on cognitive effort allocation and, critically, how it is integrated with reward in determining such allocation. To our knowledge, neither of these have been *directly* captured by previous research. We view this goal as analogous to the goal of research on the cognitive neuroscience of risky choice. Early studies developed paradigms to investigate the mechanisms by which reward magnitude and probability were integrated to determine the expected utility of different actions. This research became influential not because the behavioral findings were unexpected but rather because it had been previously unknown how risk and reward were separately processed and, more importantly, because varying these orthogonally allowed researchers to study the mechanisms by which behavior is driven by expected utility per se rather than only how it is driven by reward. By analogy, within research on motivation-control interactions, there are well-established findings linking reward and control allocation, but not demonstrating how reward is integrated with some other variable to form the equivalent expected utility for control allocation. The EVC theory builds on past theories that propose that efficacy is a central component in this integration.

2) In my view, a lot of (partly important) details were missing, which I'll try to list here:
 - the colors (and luminance!) of the cues (both in general, and concerning possible counterbalancing).

- Thank you for pointing out this oversight on our end. We now clarify that we changed the appearance of the cues also to accommodate pupillometry. We have added the following information on p 23 "We also changed the appearance of the cues as depicted in Fig. 2. We selected putatively equiluminant colors (grey: C:30.98, M: 19.61, Y: 20.78, K: 0; pink: C: 9.8, M: 42.75, Y: 0, K: 0, blue: C: 61.96, M: 0, Y: 0.39, K: 0). Luminance (computed post-hoc for the 4 cue stimuli as a whole) was similar across the individual cue stimuli (no efficacy low reward 1.4199 cd/m², no efficacy high reward: 1.3980 cd/m², high efficacy low reward: 1.3829 cd/m², high efficacy high reward: 1.3577 cd/m²) and approximately 1.4 cd/m². We used the same stimuli throughout and did not counterbalance. Note that the small deviations in luminance do not correspond to the observed patterns in pupil dilation."

- more details on how exactly participants were instructed, how the yoking procedure was described to the participants etc. (i.e., what was the participants' expectation in this regard). This might also have theoretical relevance. E.g., in experiment 1, participants behaved a bit like they didn't "get" that low efficacy meant that their performance was really entirely irrelevant. Similarly, in experiment 2, the P3b main effect of efficacy could be value-related in case participants were not sure whether reward expectation was really identical for low and high efficacy. The authors present plausible explanations for these effects, but, long story short, it would be important to know how participants were instructed in this regard and whether they believed it.

- Thank you for raising this point. We agree that it is important for the reader to know how participants were instructed. We now report the instructions in the method (p 21). Practice block instructions read as follows: "In the next block, you again need to press the key associated with the color of the text on the screen. From now on, you will have the opportunity to get an additional bonus based on how you perform the task. You will be told on each trial how performance could affect your bonus. Before each word appears, you will see an image that tells you 2 things: (1) the amount of reward you could earn and (2) whether or not your performance will determine if you get that reward. When you see one of the two images above, you can get a low (\$0.10) or high reward (\$1.00) if you respond quickly and accurately. The two images above ALSO indicate that you can get a low or high reward, BUT the gray hands indicate that your reward will have NOTHING to do with how quickly or accurately you perform. Instead these rewards will be determined randomly. As long as you provide some response on that trial, you have some possibility of getting a low (\$0.10) or high (\$1.00) reward. Although these rewards will be random, you will be just as likely to get a reward on these trials as the trials with the blue hands." In both studies the experimenters verbally confirmed that the participants understood the instructions. In study 1 the standard procedure was even to have participants explain the meaning of the cues back to the experimenter. As we unpack in the discussion, there are many established phenomena (accuracy bias, switch costs, etc.) that plausibly underlie the limited performance decline under low efficacy. In support of the switch cost interpretation, in preliminary data from a blocked version of our paradigm, we find substantially stronger efficacy-related drops in performance.

- the time-range and electrode selection for the P3b and CNV (and ERN); how exactly were those selected (avoiding circularity); or was there another approach to avoid type-1 errors that I overlooked? In particular, the (unequal) subdivision of the CNV seemed a bit odd. Unfortunately,

no such selection information was preregistered (which would have resolved any possible concerns). Maybe the effects are so strong that the specific time-ranges and electrodes do not matter, but a clear description of how these elements were selected is necessary.

- Thank you for making us aware that we did not sufficiently clarify why we selected the respective time-windows. We agree that it is important to quantify ERP components in a principled way and here selected all time windows a priori based on the literature (with the exception of the FRN, where we merely constrained the peak detection – which was performed in a condition agnostic manner (cf. Luck & Gaspelin, 2017) – based on the literature). We now added the relevant citations to each component in the method (p. 24/25). “We focused our analyses on these event-related potentials, quantified agnostic of condition with ROIs and time-windows determined a priori based on the literature⁸²: cue-locked P3b (250 to 550 ms, averaged across Pz, P3 and P4 27,⁸³), cue-locked late contingent negative variations (CNV; 1000 to 1500 ms post-cue, i.e., -500 to 0 ms pre-target, averaged across Fz, FCz, and Cz 30), response-locked correct- and error-related negativities (CRN/ERN; 0 to 100 ms 43,⁸⁴), and feedback-locked feedback-related negativity (FRN; quantified peak-to-peak at FCz as the difference between the negative peak between 250 to 350 ms and the positive peak in the preceding 100 ms from the detected peak⁸⁵).” In revising this section, we realized that we included components for exploratory analyses not referred to in the manuscript or supplement, that includes the stimulus locked N2 and the early CNV. We have now removed those from the Method as to not confuse the readers.

- some more information on the modelling approach. Specifically, I was not sure what “fixed effects” (see e.g. figure caption 3) refers to exactly, in this context...

- Thank you for raising those points. We now realize that we need clarify our terminology given the tremendous variability of what these terms refer to across different methods and disciplines. To briefly summarize, we fit linear mixed effects models (i.e., multilevel or hierarchical regression models) that simultaneously estimate coefficients for group-level effects (typically referred to as “fixed effects” in mixed effects models) and variance across subjects (typically referred to as “random effects” in mixed effects models). Fixed effects in linear mixed models are like the averaged subject-level betas in typical fMRI analyses, with the main difference and benefit being that they are estimated hierarchically and simultaneously with the random effects, which increases the robustness of the results. We have now clarified the meaning of this term in the figure caption (Fig. 4).

...Moreover, it was not clear to me how the approach of integrating the ERP data in to the behavioral analysis accounted for the condition-level differences? I might misunderstand, but it sounded to me as if all conditions were lumped together, which could end up repackaging the same condition-level effects that were already present in both modalities. I am not sure I am clear here; I am thinking of how for fMRI data one would use parametric modulators that only account for residual variance after explaining the categorical condition differences.

- Thank you also for raising the question regarding the decomposition of variance using experimental conditions and neural activity. Our approach tries to answer two questions at once: (1) do neural indicators account for variability in performance above and beyond the experimental conditions and (2) do they account for variability in performance better

than the experimental conditions? So in some sense, the very point of the analysis is to test whether we can repack the variance – i.e. do the neural indicators account for the variance previously associated with the conditions and beyond. In order to address the first question alone, we agree that the approach you suggest is a stringent test. We have run the residual approach and the effects of P3b and CNV do not change substantially if we do that. All effects remain significant. While comparison of the betas is difficult for accuracy as we move from a binomial to a continuous DV, in the accurate RT data, the betas in the new analysis are basically the same as in the simultaneous approach: P3: before -7.04 (CI: -8.87 – -5.21) now -6.99 (CI: -8.80 – -5.18), CNV: before: 15.58 (CI:12.02 – 19.15) now: 15.45 (CI: 11.94 – 18.96). This approach does not allow us to see how the estimates for the conditions change if we control for the neural correlates, however. Hence, following your approach, we also tested whether there are residual condition effects after analyzing the data as a function of the neural signals (and covariates) only. Here, like in our main analysis, we found that the main effects of incentives were present previously were reduced after controlling for neural signals. Such changes would be expected if neural activity (partially) mediated the relationship between the experimental conditions and performance.

Including all the predictors in a single model allows the predictors to compete for variance and lets us test both of these hypotheses with a single analysis, and critically, this approach yields virtually the same results as the additional stepwise analyses reported above. Given the parsimony and enhanced interpretability of our original approach, we prefer to continue to use our current (simultaneous regression) approach, but have noted that a stepwise approach yields quantitatively very similar results to alleviate potential similar concerns at the end of the readers in the Method. “In separate analyses we confirmed that similar results are obtained using a step-wise approach, analyzing residuals of the behavioral model with CNV and P3b as predictors or analyzing incentive effects on residuals from a model with ERPs but without incentive conditions, suggesting partially non-overlapping variance.” (p 26)

- relatedly, I think it would be helpful to extend a bit on the inclusion of the single-trial baselines as covariates; since this is not standard, I would appreciate a sentence or two explaining this beyond just the provided reference. It would also be good to be a bit more explicit on what main effects of, and interactions with “trial” (and not “trial number”, if I understand correctly) then mean...

- Thank you for asking this question. We used this approach because variability in the pre-trial interval systematically confounds the measure of interest. While on average the activity in the baseline is zero, at a single trial level, that is not true. That means that by subtracting the baseline, we may remove some variance between trials, but because the zero-activity baseline is only true on average, we also inject noise in that baseline into the subsequent trial. Thereby all components within that trial share that source of variance and are artificially positively correlated. Once the baseline is regressed out, this is no longer true and we “cleaned up” the induced shared variance. This approach is necessary if one is interested in multiple events within a trial. If we didn’t control for pre-target baseline effects, the signal would be confounded with the pre-target CNV effects – for example, the effects of interest would be sign-reversed if we used the pre-target baseline, or they would be in the same direction if we used the pre-cue interval. Critically, controlling for what happens during the baseline will address these sign-

reversal problems. We now added a brief explanation in the method. We now expand on the rationale on p. 27 as follows: “This approach accounts for variability prior to the effect of interest that can otherwise induce spurious effects due to noise or spill-over from previous stages of the trial. Although noise in the baseline is assumed to average to zero when using traditional ERP-averaging approaches, this assumption does not necessarily hold for single-trial analyses. To address these potential spurious effects, we follow recommendations to include the baseline as a nuisance regressor”

We further included trial – which is in fact trial-number in the analyses to account for potential learning or fatigue effects – i.e. to just capture trends over time. We do not test for interactions with trial except in the group comparison. In that case the interaction with group reflects the fact that we observed learning effects in Study 2, where performance improved across trials, but not in Study 1.

...Finally, depending on the motivation, why was this approach not extended to the cue-locked analysis of the pupil-size data?

- We did in fact run the same analyses for the pupil data as we did for the EEG data (controlling for baseline, trial etc.), however, we now take an entirely different approach to the pupil data as we explain below.

- report the range of reward rates, both concerning what was paid out, and how successful participants were (i.e., how often they would have qualified for reward). Was this comparable across most participants?

- We now report summary statistics on the proportions of rewarded trials and how often participants met the criteria in the Supplement (Table S 10). As can be seen, the reward rate matches performance closely. Study 1: Rewarded: $M = 0.77$, $SD = 0.10$; fast & accurate: $M = 0.76$, $SD = 0.10$, Study 2: Rewarded: $M = 0.61$, $SD = 0.16$; fast & accurate: $M = 0.60$; $SD = 0.15$

- how were participant informed about the (random-trial-selection) reward scheme? I assume that they were given veritable information about the random drawing of ten trials, but how exactly was this information presented?

- Participants were provided with the following instructions right before the main task: “From now on, you will continue performing the same task, but it will not be practice. Every trial can influence your ultimate bonus. At the end of the session, we will choose 10 trials at random and pay you an additional bonus based on the total amount of money you earned across those 10 trials. This means you have the opportunity to earn up to 10 additional dollars on this task.” The text of these instructions are now included in the method (p 21).

- a bit more information on which ICs were excluded would be helpful (how many? What did they seem to reflect?). I understand that this was guided by an algorithm, but more details on how it was used would be helpful.

- Thank you for your question. We now expanded on our procedure so readers have a better sense for how we analyzed our data. On p. 23/24 we now write:

“We used ICLabel, an extension made by EEGLAB’s developers (Pion-Tonachini, Kreutz-Delgado, & Makeig, 2019, NeuroImage), to identify ICs that were classified as eye or muscle ICs. The algorithm assigns probabilities to 7 categories: brain, muscle, eye, heart, line noise, channel noise, other. The extension also provides an interface (see <https://sccn.ucsd.edu/wiki/ICLabel>) that shows the topography, time course, power spectrum, and ERP-image (sorted by trial number) of each IC. Guided by ICLabel’s classification algorithm, for each participant, we excluded, on average, 2 to 3 eye frontal components (e.g., blinks, vertical/horizontal eye movements) and 1 to 3 muscle components (usually ICs that showed maximal activity at temporal sites). ICs were considered blinks or eye movement ICs and excluded if (1) there was a high probability (> 85% & < 1% brain) of them being classified as an eye-related IC *and* (2) the IC time course activity resembled blinks or vertical/horizontal eye movements (e.g., activity that looks like step-functions) *and* (3) the topography showed maximal activity at frontal sites (see <https://sccn.ucsd.edu/wiki/ICLabel> for an example of such an IC). ICs were considered as muscle ICs and excluded if (1) there was a high probability (> 95% muscle & < 1% brain) of them being classified as a muscle IC *and* (2) the power spectrum resembled noise or muscle activity more than neural activity (i.e., power peaks at higher frequencies rather than lower frequencies).”

- Fig. 4 could use some more image resolution; I would recommend larger maps (and labels to the hot-scale, even if this is a stats value, rather than uV), and larger sensor plots. It is really hard to see the CNV effect in particular. I would also endorse some low-pass filtering of the sensor data for cosmetic/visibility reasons.

- Thank you for pointing this out. The topographies were indeed quite small and the color bar needed a label. We have now revised this figure and exported it at a higher resolution.

- questionnaires are mentioned in the preregistration and the method section, but nothing beyond that (I think). I see that there were no predictions about them, so maybe they were only acquired to control for possible confounds or in the context of a larger endeavor? Please clarify (if I didn’t miss it).

- Thank you for pointing this out. We now clarify that, as the reviewer suspected, these measures were not collected with the intention of testing a specific hypothesis within the current data on their own (p. 22). We did not believe that the present sample sizes, chosen for the present purposes, justify analyses of individual differences, but that we might test predictions across datasets in the future.

- please extend your discussion of the P3b (p. 16, middle paragraph) to what you think it means in your data.

- We have now clarified and expanded on our interpretation of the P3b findings. We assumed that the P3b indexes the evaluation of the cue and used it as such an index. Because it also predicted variability in performance and has been linked to decision-making, we raised the alternative hypothesis that it may index effort decisions. However, in that same sentence, we argue that this interpretation is less likely than the original interpretation we put forward in the introduction. We have now changed this sentence to clarify this: “It is therefore possible that control allocation was already being determined

at this early stage of the trial but, in conjunction with past findings⁶⁷, it is equally or perhaps more likely that the P3b indexed the initial evaluation of the motivational relevance of cued incentives, as we originally hypothesized.” (p 16)

- lines 495, 496: please motivate this approach more, and spell out its consequences. It sounds like it might be a good idea, but intuitively, I can also see this affect the data in problematic ways, since you assume that the P3b and CNV are inherently linked.

- Thank you for asking about including the P3b as a nuisance regressor in our CNV analysis. EEG activity is strongly autocorrelated. This leads to condition-independent variance shared between components. To avoid spillover effects from the P3b to the CNV, we controlled for P3b activity in a condition-independent manner. This can be thought of like a baseline correction. We did however run complementary analyses early on to verify that this procedure does not qualitatively change the effects, which it does not.

We have now added the following explanation on p 27: “In the CNV analyses, we further controlled for variation in the preceding P3b amplitude, because here, likewise, due to the autocorrelation of the signals, larger P3b amplitudes (a positive-going ERP) would require larger subsequent CNV amplitudes (a negative-going ERP) to counteract the larger positive P3b amplitudes and reach the average levels of CNV amplitude..”

3) I don't think that this is huge, but I found the RT time-outs to be too narrow, not least because congruency is a factor here, and hence RT distributions might have been differentially cut off. As for the 750-ms deadline, I assume that it wasn't adjusted during the experiment (hence my question above for more details), right? If so, is it not possible that some participants found the task quite easy and others very challenging? Not a big concern given the within-subject nature of the experiment, but could have introduced an extra variance source...

- Thank you for raising this concern. It is true that the perceived difficulty probably varied between participants. As with most cognitive control tasks, incentivized or not, differences in perceived difficulty will likely induce variability in the incentive effects. Something we should clarify, though, is that 750 ms was the (performance-based) reward deadline, but participants had up to 1000 ms to respond. It was relatively rare for participants to miss this 1s time-out - across all conditions and participants, only 2% of trials were missed in Study 1 and 8% in Study 2. It is true that incongruent trials were cut off more frequently than congruent and neutral trials, however, congruency was varied independently of incentives, not the focus of the present work, and controlled for in the analyses. Moreover, if the task had not been challenging, there would have been little or no room for incentives to improve performance.

4) I have significant problems with the pupil data. Beyond the surprising result pattern (as acknowledged), effects start way too early, to an extent that is implausible. Typical “psychological” pupil effects don't arise until close to 1 second (maybe 600 ms), and even the light reflex takes a bit of time. Hence, effects at 40 ms (and 220 ms) post-cue are simply implausible as event-related responses to those cues, and probably point to a problem with response overlap from the preceding trials (and hence to a non-random trial sequence). Or do I misunderstand the timing information here? I don't know whether the inclusion of the single-trial

baseline activity should attenuate such problems or might exacerbate them (then again, this seems to have been applied only to the target and feedback screens), but as they stand, I don't think that the pupil data are really interpretable. I am sorry for the strong statement, and will happily revise it if the authors can point me to a point of misunderstanding, of course. [One thing the authors could try as a control analysis would be to look at target-locked congruency effects, which are well-documented in the literature as starting around 800 ms post-target]. If taken at face value, more discussion would be needed about what these results likely reflect specifically (rather than mostly what they do not reflect).

- We are grateful to you for identifying and raising this important concern. We entirely agree that these early onsets are implausible and share your concerns about the issues these pose for interpreting the subsequent peaks in those pupillary responses. Your comments prompted us to follow up with more targeted analyses aimed at characterizing these pupillary responses in their entirety, and isolating the influence of different components of our task design on those responses. In brief, we've identified the cause of these early onsets in the relatively short ITIs (0.5-1.0s), which we had optimized for EEG but were less ideal for disentangling pupillary responses. As a result, feedback-related pupillary responses spilled over from this jittered period into the early periods of the cue-locked pupillary response, spuriously producing the early onsets you highlighted. To test and control for such response overlap, we adapted a deconvolution analysis approach similar to what is done for fMRI analyses, modeling event-related pupillary response functions just as one would BOLD hemodynamic response functions. Using this approach, we were able to dissociate the cue response from the overlapping feedback response from the previous trial (as well as target and response related responses on the current trial) and estimate the effects specific to that event. When controlling for response overlap, we continue to find our previously reported effect of efficacy on (lower) pupil diameter but no longer observe a significant effect of reward (see Figure below). Thus, even after removing variance attributable to earlier task stages, we continue to see evidence that higher expected efficacy was associated with lower rather than higher arousal. (As a proof of principle for this deconvolution approach, we separately confirmed that we were able replicate the well-documented target-locked congruency effects, i.e., incongruent trials have larger pupil responses than congruent trials, $p < .001$)

We have now replaced our previous pupillary analyses with these new ones that address the overlap issues identified by the reviewer, while also highlighting the suboptimality of our task design for this specific measure. Given this general concern, and the fact that the pupillary findings were ancillary to our main hypotheses, we have moved this finding to the supplement and modified our interpretations appropriately. These changes do not impact any of our main conclusions regarding our behavioral or EEG findings.

5) I had the impression that the authors were struggling with this quite a bit themselves. I have to honestly say that I am not sure what contribution of the rather elaborate supplemental material about the ERN, FRN, and fronto-central theta power can make. This relates to the likely dependence of response-locked and feedback-locked data (which has to differ as a function of efficacy, as the authors acknowledge), and, maybe even more importantly, again as identified by the authors themselves, to the fact that the cue-target set-up pretty much abolishes the possibility for classic reactive control processes. I am not sure what to suggest here, but I would consider further shortening those elements, certainly in the main manuscript.

- Thank you for your candor. The rationale for reporting these findings in the supplement was as follows: First, we wanted to transparently report the results for all preregistered hypotheses at least in the Supplement. Second, we actually consider the ERN and FRN findings interesting novel results – both in their own rights and in their relationship to motivation and effort. A straightforward prediction – our preregistered one, in fact – consistent with a substantial body of work is that motivation amplifies response- and feedback-based performance monitoring. Our findings are not consistent with this prediction. They also contrast with previous findings in which – as it turns out – motivation was indexed differently (e.g. difficulty with all the issues we unpack in the main text) or feedback served a different purpose. We thus unpack why that might be and shed light on the functional significance of these signals in our paradigm in contrast to other contexts. The interplay between internal and external monitoring has not received as much attention as it deserves and effects of predictability on reward processing are often overlooked and confound e.g. the interpretations of effort effects on neural reward processing. As we discuss, the literature that explicitly investigates feedback reliability typically uses tasks in which prediction errors serve learning rather than reward processing in its own right, and that leads to very different effects on feedback processing, where people downweigh unreliable feedback and reward effects are attenuated. Here we point out that when feedback is not relevant to updating response-outcome relationships, and outcomes can be predicted based on performance, reward effects are muted rather than enhanced. That suggests that the effect of reliability depends on what the feedback is used for! This functional dependence complicates the interpretation of reward effects in many ERP (or any neural) studies,

and is not discussed at all. Only recently, the interplay between internal and external performance monitoring has received more attention (cf. Frömer, Nassar, Bruckner, Stürmer, Sommer & Yeung, 2020, de Gee, Correa, Weaver, Donner, & van Gaal, 2020). It is also novel, that the ERN not only integrates information about the accuracy of the given response, but also its timing relative to the deadline, and a condition-dependent reward prediction error like signal. This suggests that it is really goal-achievement that is being evaluated in a predictive manner, not response accuracy per se or that these are mere protracted conflict effects. We agree that unlike those (in our minds pretty exciting) monitoring effects, the effects of reward and efficacy on the ERN are somewhat confusing. We have not resolved this, but it is likely that they arose as a function of dynamic control/performance evaluation. This hypothesis will need to be tested in a study specifically designed to do so.

Following your recommendation we have now, however shortened the ERN section in the main text.

Minor

1) The rationale for not having a fixation dot was unclear to me (lines 433/434). An offset response (of the cues) will be more problematic here than would have been the additional onset of a fixation cross, in my opinion.

- Thank you for raising this point. We now realize that our description was ambiguous. We did not remove the fixation-cross and show a blank instead (see also Figure 2 for the events on each trial), but transition directly from the cue to the target, rather than having an additional stimulation in between those two. The counterfactual would result in the offset/onset visual evoked potentials to the fixation cross during the measurement of the CNV and in the target pre-stimulus interval. This way, we avoided or eliminated the visual evoked potential complex, which would have contaminated the main component of interest (i.e., CNV). We now clarify this on p. 22/23: “No fixation cross was presented during the cue-target interval; that is, the cue transitioned directly to the target to avoid inducing visual evoked potentials that would influence the amplitude of the CNV...”

Reviewer #2 (Remarks to the Author):

This manuscript by Frömer and colleagues presents results from a behavioral experiment and an EEG/pupillometry study investigating the effects of efficacy and expected reward on the investment of effort or, more generally, the recruitment of cognitive control. Using a Stroop task modified in a clever way to specifically manipulate efficacy while keeping difficulty and expected reward constant the authors show that people invest more control when this control is more rewarding and more efficacious (this is mainly reflected in RTs for correct trials). They furthermore show that both, expected reward and efficacy, modulate the P3b to the cue indicating these values and the prestimulus CNV. The CNV amplitude also reflects the interaction of both parameters. In line with the Expected Value of Control Model people appear to integrate worth and worthwhileness to determine the amount of to-be-recruited control. In addition, the authors present interesting results for performance-monitoring-related ERPs.

This is a very nice study using an elegant design. The methods are sound and state of the art and the results are convincing. The paper reads well and is easy to follow. The results advance the field of cognitive control research and are highly interesting even beyond this field. I have only a few minor comments:

1. It's just about wording/semantics: the paper seems to suggest that effort and recruitment of cognitive control have an equal meaning. Is that always the case?

- There are of course active debates surrounding the nature of mental effort and its relationship to cognitive control. One of these relates to whether the underpinnings of mental effort are cognitive control processes or something else (or something in addition). Based on prior work (elaborated on, for instance, in Shenhav et al., 2017 ARN), we rely on the assumption that mental effort is indeed underpinned specifically by cognitive control. A second question, that is perhaps closer to what the reviewer is asking, is whether this effort is always *costly* or whether it can also be *rewarding*. This is an even less settled question than the first one, and one that we have also examined closely (e.g., Inzlicht, Shenhav, Olivola, 2018, TiCS). In short, we think that there is compelling evidence to suggest that effort is not only costly, but that there may at least be an additional reward function that scales with these efforts. However, this is still consistent with our proposal that mental effort being driven by cognitive control, only not with the notion that effort (mental or physical) is something we only find aversive. Importantly, whatever the underlying reward/cost functions on effort/control, these are being held constant while we vary expected reward and efficacy (which are tied to effort's payoff rather than effort itself, as illustrated in Fig. 1).

2. What does the significant baseline effect for the CNV (Table 1) mean?

- This is a good question. In these analyses we actually controlled for the baseline twice: We analyze our ERPs residualized on the baseline and then still include the baseline in the model. For the P3b that means that there's no effect of baseline (whereas there would be for the non-residualized one). If we don't include the P3b in the CNV analysis, we similarly do not get a significant baseline effect. However, when we do include it (also residualized) that seems to soak up shared variability due to autocorrelation and leaves remaining variance to be explained by the baseline. So, what this could reflect – with the caveat of this being pure speculation – is that the more participants already anticipated

the cue (which would lead to a CNV-like signal in the pre-cue interval), the less they further recruit resources prior to the target, possibly because those resources are already allocated. An alternative, less interesting, but more likely possibility is that initially residualizing the CNV on the baseline is less effective because more variability is induced over time (cf. Alday, 2019 showing that the correlation with baseline decreases over time). As we control for one of the sources of additional variability (P3b), the effective weight of the baseline increases again. Luckily, neither the P3b nor the baseline effect are of interest and merely serve as controls. The effects of the remaining variables do not change qualitatively if we remove these variables.

3. I did not fully understand the reasons for the statement in lines 337 and following that the pupil responses more closely track arousal than proactive control. More specifically, this would mean that lower efficacy (increasing pupil size) is associated with higher arousal. Why should this be the case?

- We have now clarified this interpretation in the results and discussion sections. The way we thought about it was that our results parallel other effects of uncertainty (e.g. in Matt Nassar's work) that have been interpreted as arousal effects. In our case, outcome uncertainty is much higher for low than for high efficacy, because participants cannot reduce outcome uncertainty on that trial by how hard they are trying. "Although this pattern was not predicted in advance, these findings are consistent with the interpretation that pupil responses in our paradigm track arousal – induced by higher uncertainty under low efficacy – instead of proactive control^{39,42}." (p 17)

4. The supplementary ERN/FRN analyses are very interesting. I just wonder whether for the analyses shown in Fig S2E there were enough trials in each subject, in particular for congruent and neutral trials. How reliable is this analysis? (it is cool though)

- These effects were estimated collapsing across correct and incorrect trials (there was no interaction with accuracy). So this is not dependent on the error distributions. We also use a hierarchical approach that leverages all participant's data to estimate this effect, regardless of the specific number of trials. That makes this a lot more robust than, say a two-step approach (within subject regression and then test on those betas). It would definitely be nice to follow up on these trial dynamic effects on the ERN in a study that is specifically designed to do so. The prediction to test here would be that the information that drives variability in the ERN over time is the information that drove the corresponding response. For faster responses, incongruent information may be salient and elicit an ERN when the response is correct or when it is incorrect, whereas when the response is not driven by the incongruent information, it is less likely to result in a fast response and the later the response occurs, the more likely performance is evaluated as negative because it's not going to meet the deadline. So to test whether performance evaluation signals depend on the relevant information available/salient at that time, one would need to experimentally manipulate the temporal dynamics of available information in the trial and then analyze response-locked signals as a function of that.

5. A more general question out of curiosity that may not be answered by this study and touches metacognition rather than the focus of this study: Can people perceive the difference of efficacy and difficulty, in particular, if there was no cue (or the meaning of the cue would have to be learned)?

- That is a super interesting question and touches on credit assignment. Long story short: they ought to be able to learn that (albeit possibly very slowly) and that should also require the interplay between internal and external monitoring. In a recent study, we showed that when trial-by-trial feedback about efficacy (performance-based vs random outcome) is provided, participants are able to integrate that feedback to track changing efficacy levels and use these estimates for control adjustments (Grahek, Froemer, & Shenhav, in prep). How participants disambiguate difficulty and efficacy contributions to their performance outcomes and whether they are able to do so, remains an open question that we plan to address in future work.

Reviewer #3 (Remarks to the Author):

This manuscript reports the results from two experiments, the second pre-registered on the basis of the first, which were designed to test the expected value of control (EVC) hypothesis as it relates to both behavior and neurophysiological readouts as elicited by a novel task. The subject matter of this manuscript is of theoretical, and potentially also practical, significance. The methodology is sound and rigorous. The results are interesting, at times surprising, and always-clearly described.

Please find my suggestions below.

Suggestions related to theoretical clarity: As written the manuscript's theoretical foundations may be somewhat opaque to the broad readership of this journal. For the below confusions, the authors may wish to rephrase their existing language:

(1) The authors show evidence that correct responses are faster for high than low efficacy trials, which is said to be consistent with EVC, but in the supplement it is mentioned that incorrect responses are actually faster for low efficacy than high efficacy trials. The latter is interpreted as "consistent with effort minimization by, e.g., responding with any key." If the latter interpretation were correct for low-efficacy trials, presumably one would then predict that correct responses would also be faster in low-efficacy trials. Can the authors please clarify this apparent contradiction?

- Thank you for this important question. We can definitely see where our (admittedly post hoc) interpretations of error trial RTs (fastest for low-efficacy) can seem to conflict with our a priori predictions regarding correct trial RTs (fastest for high-efficacy). We think there may be a few different factors that can help resolve this apparent conflict. First, as we have recently shown in other work, people simultaneously weigh multiple different control strategies and engage these strategies to different degrees based on what is most adaptive for one's current incentives (Leng et al., 2020, BioRxiv). For instance, in that study we show that people vary in the extent to which they engage attentional processes (that support fast and accurate responses) versus adjust their response threshold (i.e., their speed-accuracy tradeoff). For the current study, the former mechanism is generally most adaptive, and the bulk of our findings can be accounted for by variability in such attentional processes (formalized as the rate of evidence accumulation in our model). But, while our current task wasn't optimized to detect this, we speculate that people may have additionally been weighing a threshold-based

strategy for deciding whether to engage in further processing a given stimulus (versus respond randomly). In particular, since the vast majority of errors are incongruent trials, it is possible that on a subset of these trials participants encountered the incongruent stimulus and defaulted to a random responding strategy rather than processing the stimulus further.

Specifically (and again we emphasize that this is only speculation at this point), participants may have anticipated a neutral or congruent trial, as either of those was more likely than an incongruent trial, and allocated resources accordingly. Once the incongruent trial occurs, there is a mismatch between the control that was allocated and the control that is required to get it right. At this point, there is room to update the cost benefit computation. Given the new information, trying to get a low reward low efficacy trial right may just not be worth it and so the error is accepted and a fast response produced to not waste time and effort on a low value trial (incongruent trials are much more likely to result in either errors or time-outs). Producing ANY response is the minimum requirement for obtaining a reward, when efficacy is low. So when the probability of giving an accurate response by increasing control is low, reducing response caution to respond quickly should be the optimal thing to do. On the other incentive trials, this computation may be coming out more heterogeneously, such that participants attempt to be correct but fail, or merely take longer to decide whether to try and be correct and don't submit a Stroop response in the meantime before eventually submitting a random response. We now think that these dynamics are important to consider, and are what leads to changing predictions over time (as information changes) and for different outcomes. While we initially looked at the error RT data merely for completeness and to complement the ERN analyses, they have revealed this intriguing pattern that motivates future studies to test behavioral and neural predictions for a dynamic updating of EVC account. To address your concern we have now added the following to the respective section which is now in the supplement following Reviewer 1's suggestion: "On the large proportion of incongruent error trials participants may further have reevaluated the value of control in a reactive manner given the new information about the difficulty of the trial. Future research will need to test specific predictions of this ad-hoc interpretation in a principled manner." In the main text we now write: "Follow-up analyses suggest that this pattern may result from different dynamics in control and response evaluation between conditions (see Supplement, Fig. S3; Table S5 -7)." (p 13)

(2) The authors demonstrate that reward and efficacy similarly interact for both behavior and the CNV, and moreover that (when included in the model) differences in the CNV continue to predict behavior even when accounting for the effects of reward, efficacy, and their interaction. However, it appears the converse is also true - efficacy continues to predict accuracy and correct RTs above and beyond the effect of CNV; similarly, the reward:efficacy interaction explains at least correct RTs above and beyond the effects of CNV. Can the authors please clarify if (or how) this may challenge the interpretation of CNV "indexing" the integration of efficacy and reward, as predicted by EVC?

- This, too is an excellent point and we agree that a straight forward prediction would be a full mediation of the incentive effects by the CNV. We were somewhat surprised not to find this and also that P3 predicted behavior as well. It is likely that we didn't find a full mediation by CNV because of the noisy nature of the neural signals (we also use a pretty large time window for CNV quantification, e.g. when instead using only the

terminal 200 ms, when CNV is maximal, we capture more variance otherwise associated with incentives and their interaction). While this may be part of the explanation, we believe that the remaining incentive effects have to do with the point raised in (1). The anticipated demands change over time as participants learn about the stimulus and its congruency and so should the required level of control to get it right. Thus, there is additional variability in control allocation as a function of incentives that cannot possibly be captured by the CNV because at the time of the CNV, this update decision has not been made/ the requirements are not fully known. As for the CNV effects, we believe that the CNV reflects control allocation. This control allocation should be partly determined by the incentives we offer, but not entirely. Motivation may in addition fluctuate from trial to trial. Participants may weigh efficacy or reward differently – such variability would lead to variability in incentive effects that can be captured by the CNV if it reflects corresponding control allocation, as we assume. Some people may even care more about accuracy than about any of the incentives and that, too should then lead to CNV effects that are independent of the incentives themselves. Thus, that incentives and CNV share some but not all variance is absolutely expected, even beyond the fact that the CNV is a noisy measure and thus unlikely to soak up all incentive variability.

We now expand on this on p. 16/17: “It is further of note that even both ERP components combined did not fully mediate incentive effects on performance. This could be due to those ERPs being noisy indicators of the underlying processes, or to dynamics following target onset that lead to additional variance as a function of incentives that cannot be explained with proactive control. Specific predictions of the latter account could be tested in future work explicitly designed to do so.”

(3) EVC would seem to make predictions about individual differences, but these are not the subject of any analysis here. In addition, the preregistration includes the collection of questionnaire data, which was presumably collected for this kind of reason. Are there systematic subject effects here that could complicate the use of a purely random effects account of subject-level variation?

- Thank you for raising this interesting point. While our model does indeed predict ways in which individual differences might modulate our incentive effects (cf. Grahek et al., 2019; Musslick, Cohen, & Shenhav, 2019), we believe that our current sample is not large enough to support robust individual differences analyses. As outlined above, we believe that factors beyond the incentives will influence effort allocation and so should differences in how incentives are valued. We are collecting these data in all of our studies to explore the impact of individual differences in aggregate data across studies. While these individual differences are interesting in their own right, and we are investigating them in adjacent lines of work, we do not think that their impact interferes with our work more than any other experimental study that does not explain all residual variance. We do now explicitly state that the questionnaires are not analyzed in the present study.

Suggestions related to methodological approach:

- Given the methodological care of the manuscript it is surprising the authors do not report any measure of reliability. Can the authors please provide some estimate of reliability (e.g., split-half or odd/even)? It is acknowledged that due to the rolling window "yoking" procedure, these reliability estimates may be upwardly-biased.

- Thank you for the excellent suggestion. We have now computed split half reliabilities for our 4 key measures of interest and report them when we first report results on those measures. For each participant and incentive condition, we computed the average measure for odd and even trials and then we correlated these. Below we're visualizing the consistency in these measures. The correlations are $r = .82$ for accuracy, $r = .91$ for accurate RT, $r = .86$ for P3b and $r = .75$ for CNV.

- The authors appear to use a mean-centered explanatory variable for trial number to account for linear practice effects. However, it's not expected that practice effects are linear. And, the functional form of practice effects will be highly relevant for building on this work (e.g., in defining block lengths and total number of blocks). Can the authors please address this point, for example adding a supplementary graphic plotting the residual RTs (from a model that does not include trial length as an explanatory variable) so that future work in this area can be informed by the practice effects that may or may not be captured by linear effects of trial #?

- This is a great point. Using the approach suggested by the reviewer, we have now examined the shape of the practice effects that we observed in Study 2 (no such effects were observed in Study 1). As shown below, residual RTs appear to show a trend that is well captured by a linear component. In the plot we use an unconstrained spline fit that yields an approximately linear effect with a slightly faster decrease early on, followed by further, but slower decrease. We now included the below figure in the supplement.

- The authors allude to how they determined the random effects structure for their model (using "singular value decomposition", by which they presumably mean they added random effects only until the models could no longer be estimated). Can the authors please provide the final models they used, including random effects specifications? It would also be worth confirming to the reader that these models meet the relevant assumptions, that Q-Q plots for the models had been inspected, that there was no problematic collinearity between predictors, etc.

- Thank you for raising this point. We adopted this approach based on recommendations laid out in a recent paper by Matuschek and colleagues (reference below). Singular value decomposition requires fitting random effects and assessing the variance explained by each component that was fit. When a variance component is 0 that means that the model is degenerate, that there is no variance that can be captured by a given variance component. That makes the parameter estimation unstable and decreases power. Therefore, we exclude variance components that lead to such model degeneration and only include variance components that actually explain variance. We now point interested readers to the code that has the final models we fit in the method.

Matuschek, H., Kliegl, R., Vasishth, S., Baayen, H., & Bates, D. (2017). Balancing Type I error and power in linear mixed models. *Journal of Memory and Language*, 94, 305-315. doi:10.1016/j.jml.2017.01.001.

- Relatedly, there are many ways in which this data could trigger a variety of illuminating work, but I note that the authors plan to make the data available upon request. Since they have gone to the trouble of preregistering their work on OSF, I would strongly suggest the authors consider making their data available via the same mechanism. This is likely to increase the impact of the work significantly.

- Thank you for your suggestion. We agree and made the data available through OSF: osf.io/35akg. (See data availability statement p 28)

Minor points:

- Unless I missed it, the authors do not describe the procedure for creating integrated accuracy/rt z-scores. It is obvious enough how this must have been done, but it should be described explicitly in the manuscript.

- Thank you for raising this point. We should be clear that z-scores were only used for illustration, data analyses were conducted on untransformed data, as necessary for the single-trial hierarchical approach (otherwise random intercepts per subject cannot be computed). For the data shown in Fig. 3, we simply z-scored RTs across accurate trials within each participant.

- The authors do not report statistics for the interaction of efficacy and reward in the left column of Table 2 (for predictions of accuracy). Can this be included in the table or a sentence added to the note explaining why it is not reported?

- Thank you for pointing out this oversight on our end. We had previously removed non-significant interaction terms from the models, but ended up reporting the interaction in the behavioral models because it is theoretically relevant. This change had not been propagated to the neural model. We have now added this.

- suggest rephrasing "both P3b and CNV were associated with better Stroop task performance on the upcoming trial" to "both P3b and CNV were associated with better Stroop task performance when the target appeared" or similar. I was briefly led astray in thinking that there might have been an analysis of n+1 trials in the manuscript.

- Thank you for pointing out that we might garden-path readers that way. We have now replaced this phrase in multiple places in the manuscript.

- the correct citations for lme4 and lmerTest are as follows.

Kuznetsova A, Brockhoff PB, Christensen RHB (2017). "lmerTest Package: Tests in Linear Mixed Effects Models." *Journal of Statistical Software*, 82(13), 1–26. doi: 10.18637/jss.v082.i13.
Bates D, Mächler M, Bolker B, Walker S (2015). "Fitting Linear Mixed-Effects Models Using lme4." *Journal of Statistical Software*, 67(1), 1–48. doi: 10.18637/jss.v067.i01.

- Thank you for pointing out this oversight on our end. We have added the Kuznetsova et al reference, and have made sure our Bates et al. reference contains all of the information in the one you provided (to the extent allowable by the current citation format).

- At one point in the introduction the authors refer to the importance of "variability in efficacy (holding expected reward and difficulty constant)", but I believe that what is meant here is "differences in efficacy." Variability raised questions for me about the distribution of expected reward and difficulty, which is relevant to the definition of optimality (it's unclear why one may want to maximize the expected value as opposed to minimizing the lower bound; etc). Readers may better avoid this garden path if you use the simpler word "differences."

- Thank you again for pointing out the potential for garden-pathing readers. We have now replaced variability with difference as you suggested.

Reviewer #1 (Remarks to the Author):

The authors have thoroughly revised their manuscript, with many helpful additions and clarifications. Many of my concerns have been resolved. I was particularly glad to see that the implausible pupillometry results were addressed, although I will admit that now it is not immediately clear how well the new approach actually fit the data. Still, their role has been appropriately downgraded, which I find sufficient either way.

I still have a couple of elements that I would like to bring up, however. The first is conceptual. Comments 2-4 are all methods-related, but somewhat outside of my core area of expertise, so I only bring them up for the authors' consideration. If they don't share my concerns, or do not find my suggestions helpful, they can simply ignore them:

1) The general expectedness of the results is still a bit of a concern to me, and again largely in a fashion that is based on the simple all-or-nothing manipulation of efficacy. To me, the present manipulation seems to map on earlier conceptualizations (and demonstrations) of similar phenomena (albeit using different labels; active/passive, instrumental/non-instrumental etc.). I did not run a comprehensive search, and I might misremember the details, but see e.g. here: <https://pubmed.ncbi.nlm.nih.gov/15134646/>
<https://pubmed.ncbi.nlm.nih.gov/17140674/>
Similarly, on a theory level, I feel that e.g. the motivational intensity theory makes very similar predictions, which is only dealt with in passing. As one way of enhancing the novelty of the present work, would it maybe be possible to add the new behavioral data with a more graded efficacy manipulation to the present manuscript? I got quite excited when I saw that additional data was acquired, which turned into (minor) disappointment when this data did not make it into the present manuscript. I think adding this here would make the present manuscript more impactful, and would better justify the language in the manuscript that, in places, still seems to imply finer gradations than 0 vs. 1.

2) For the brain-behavior correlations, I appreciate the clarification. Yet, for such analyses, the nuts and bolts are often a bit opaque (at least for the average reader), e.g. how exactly shared variance is dealt with, whether predictors are orthogonalized etc. Would it be possible to show the main relationships also within conditions? I think that would be the clearest and most intuitive way to make sure that there is a relationship beyond just RTs and EEG data coming from the same conditions, for which the differences were already shown by categorical tests.

3) About regressing out baselines: I think some clarification would be helpful what you mean by e.g. "Although noise in the baseline is assumed to average to zero when using traditional ERP-averaging approaches, this assumption does not necessarily hold for single-trial analyses." If I understand it correctly, the point is more about the assumption than the fact that the baseline mathematically can of course be set to zero even on a single trial. I suspect that readers could misunderstand this. Please clarify, and explain (e.g., with an example) which factors could violate the "neutrality assumption" on a trial level.

4) Concerning regressing out different EEG effects to avoid effects of auto-correlation, which you liken to a baseline correction: I might misunderstand the exact nature of what is happening, but the analysis of course has a certain "black box" flavor to it. What would happen, e.g. if the P3b and CNV meaningfully depended on each other? Wouldn't your approach then take out a relevant effect? And on the opposite side of this spectrum, since you likened this to baseline correction, and not least because the P3b and the CNV have opposite polarity, could your approach not introduce an artifactual effect from the P3b into the CNV (by "subtracting" an inverted difference "into" another time-window)? Again, I don't fully comprehend this, so this is more for you to reconsider whether you are 100% sure that your approach is appropriate, and if you are, that's fine.

5) Minor, but: "All analysis code for reproducing the reported results can be found on OSF: osf.io/35akg": is it not rather posted on <https://osf.io/xuwn9/>? Moreover, although I could have simply overlooked this, I don't immediately see all the analysis scripts that I would expect. Maybe check, because this open-science angle is of course much appreciated.

Reviewer #2 (Remarks to the Author):

The authors have nicely addressed the reviewers' points and the revised manuscript has gained in clarity.

Again, I really like the study and think that the manuscript is ready for publication.

Signed: Markus Ullsperger

Reviewer #3 (Remarks to the Author):

The authors have thoroughly addressed my comments and, I believe, significantly improved the potential impact of the paper. I would still recommend they include in the supplement some indication of the reliability of the results (not merely doing so in the response to reviews, as they appear to have done), but otherwise I have no further comments.

Reviewer #1 (Remarks to the Author):

The authors have thoroughly revised their manuscript, with many helpful additions and clarifications. Many of my concerns have been resolved. I was particularly glad to see that the implausible pupillometry results were addressed, although I will admit that now it is not immediately clear how well the new approach actually fit the data. Still, their role has been appropriately downgraded, which I find sufficient either way.

I still have a couple of elements that I would like to bring up, however. The first is conceptual. Comments 2-4 are all methods-related, but somewhat outside of my core area of expertise, so I only bring them up for the authors' consideration. If they don't share my concerns, or do not find my suggestions helpful, they can simply ignore them:

1) The general expectedness of the results is still a bit of a concern to me, and again largely in a fashion that is based on the simple all-or-nothing manipulation of efficacy. To me, the present manipulation seems to map on earlier conceptualizations (and demonstrations) of similar phenomena (albeit using different labels; active/passive, instrumental/non-instrumental etc.). I did not run a comprehensive search, and I might misremember the details, but see e.g. here:

<https://pubmed.ncbi.nlm.nih.gov/15134646/>

<https://pubmed.ncbi.nlm.nih.gov/17140674/>

Similarly, on a theory level, I feel that e.g. the motivational intensity theory makes very similar predictions, which is only dealt with in passing. As one way of enhancing the novelty of the present work, would it maybe be possible to add the new behavioral data with a more graded efficacy manipulation to the present manuscript? I got quite excited when I saw that additional data was acquired, which turned into (minor) disappointment when this data did not make it into the present manuscript. I think adding this here would make the present manuscript more impactful, and would better justify the language in the manuscript that, in places, still seems to imply finer gradations than 0 vs. 1.

- Thank you for your suggestion. We agree that the new data makes a nice contribution and have now included it in the manuscript. Note that this includes 11 subjects from this task version who were inadvertently omitted from the results presented in our previous response letter (which was based on an earlier version of our data table). With this larger sample (N = 35), we see even stronger effects of reward, efficacy, and their interaction. Below is our updated Figure 1 with these data included:

- We also now include the additional citations you suggested (Zink et al. and Bjork & Hommer), alongside others we had previously cited by Manohar and colleagues (2017) and Kohli and colleagues (2018) that are perhaps more closely related to our current study. Similar to these other studies, Zink et al. and Bjork & Hommer only look at the relationship between reward and simple motoric responses, which prevents generalization to the domain of cognitive control that is at the focus of our experiment. Another important distinction is that our task *always* requires a response in order to obtain a reward, but when efficacy is low,

this response can be incorrect or slow. With respect to other theories, it is not entirely clear to us that motivational intensity theory would predict variability in effort when reward and task difficulty are held constant and only efficacy varies (something we discuss at greater length elsewhere; Grahek, Frömer, & Shenhav, 2020, *bioRxiv*), but we do note that our theory shares the prediction of reward-efficacy integration with others such as Expectancy Theory (which the EVC theory in part builds on). In addition to predicting that these two variables will be integrated, EVC also proposes that incentive evaluation and integration constitute separate stages and we provide evidence consistent with this. We hope that these novel findings, alongside the additional findings we have now added demonstrating finer gradations in reward and efficacy-based control adjustments (Study 3), convince you of the novelty and potential impact of our research.

2) For the brain-behavior correlations, I appreciate the clarification. Yet, for such analyses, the nuts and bolts are often a bit opaque (at least for the average reader), e.g. how exactly shared variance is dealt with, whether predictors are orthogonalized etc. Would it be possible to show the main relationships also within conditions? I think that would be the clearest and most intuitive way to make sure that there is a relationship beyond just RTs and EEG data coming from the same conditions, for which the differences were already shown by categorical tests.

- To unambiguously demonstrate that the neural data does capture the same variance as the behavioral data through manipulation of the conditions, we have now conducted complementary analyses in which we nest the neural activity within conditions. We report these results in the Supplement (Table S5). In short: we find significant effects of our ERPs in the consistent direction in all conditions in accurate RT. For accuracy, the CNV findings are more consistent than the P3b findings with the betas in the same direction in all 4 incentive conditions (and significant in 3 of the 4). The P3b effects are in the expected direction and significant in both high efficacy conditions, and we find a beta in the unexpected direction (small and non-significant) only in 1 of the 4 conditions. Note that we find this overwhelmingly consistent pattern despite a significant loss of power in this analysis.

3) About regressing out baselines: I think some clarification would be helpful what you mean by e.g. “Although noise in the baseline is assumed to average to zero when using traditional ERP-averaging approaches, this assumption does not necessarily hold for single-trial analyses.” If I understand it correctly, the point is more about the assumption than the fact that the baseline mathematically can of course be set to zero even on a single trial. I suspect that readers could misunderstand this. Please clarify, and explain (e.g., with an example) which factors could violate the “neutrality assumption” on a trial level.

- Thank you for pointing out this potential source of confusion. We have now clarified that traditional analyses assume that baseline activity does not vary systematically across time points within the baseline or between trials of different conditions. That is, the signal is expected to be constant with some noise that averages out across time-points and trials. If that is true, subtracting the average baseline will lead to zero activity at time zero, and empirically relevant activity only after. This assumption can be violated, however, when activity during the baseline time-window is non-stationary (e.g., drifting over time) or when there is, for instance, a bout of noise pushing the average activity in one or the other direction relative to the event of interest. We can still theoretically set the mean activity in the baseline to zero, but this would systematically shift the time series beyond “zero activity at time zero”. This systematic shift then affects all subsequent signals in the time series and all ERPs would become positively correlated through the shared baseline. By including a regressor for any such variance across trials, we account for it. We now include examples as you recommend to clarify the issue: “Although noise in the baseline is assumed to average to zero (across time points, as well as trials) when using traditional ERP-averaging approaches, this assumption does not necessarily hold for single-trial analyses, where a non-stationary baseline or unevenly distributed noise can easily lead to systematic biases in the subsequent time-series.” (p. 27)

4) Concerning regressing out different EEG effects to avoid effects of auto-correlation, which you liken to a baseline correction: I might misunderstand the exact nature of what is happening, but the analysis of course has a certain “black box” flavor to it. What would happen, e.g. if the P3b and CNV meaningfully depended on each other? Wouldn't your approach then take out a relevant effect? And on the opposite side of this spectrum, since you likened this to baseline correction, and not least because the P3b and the CNV have opposite polarity, could your approach not introduce an artifactual effect from the P3b into the CNV (by “subtracting” an inverted difference “into” another time-window)? Again, I don't fully comprehend this, so this is more for you to reconsider whether you are 100% sure that your approach is appropriate, and if you are, that's fine.

- Adding the P3b as a nuisance regressor can indeed be considered conservative, but will not induce an inverted effect (unlike the subtraction approach, which can invert effects). If there is shared variance between CNV and P3b, our approach will account for it (through a negative correlation in your example). In the worst case, we might not find an effect that is explained away by the P3b. That would, however, mean that the CNV adds nothing that isn't already in the P3b, or to put it in cognitive terms, effort allocation would not differ from cue evaluation. As we do think that these signals index different processes and that their relationship with incentives should be different, we think that accounting for the P3b in the CNV analysis is fair and critical test. If there is no such shared variance, nothing will happen. That is the beauty of the approach and nicely explained in Alday (2019, <https://doi.org/10.1111/psyp.13451>). Thus we are in fact 100% sure that this approach is appropriate – if anything, it's conservative. As it turns out in our data, the P3b does not explain away the effects we see on the CNV and that reassures us that the CNV makes a unique contribution to the processes under investigation.

5) Minor, but: “All analysis code for reproducing the reported results can be found on OSF: osf.io/35akg”: is it not rather posted on <https://osf.io/xuwn9/>? Moreover, although I could have simply overlooked this, I don't immediately see all the analysis scripts that I would expect. Maybe check, because this open-science angle is of course much appreciated.

- Please excuse this oversight on our end. We linked to the preregistration instead of the project page (osf.io/xuwn9). Thank you for correcting this mistake.

Reviewer #2 (Remarks to the Author):

The authors have nicely addressed the reviewers' points and the revised manuscript has gained in clarity. Again, I really like the study and think that the manuscript is ready for publication.
Signed: Markus Ullsperger

- Thank you!

Reviewer #3 (Remarks to the Author):

The authors have thoroughly addressed my comments and, I believe, significantly improved the potential impact of the paper. I would still recommend they include in the supplement some indication of the reliability of the results (not merely doing so in the response to reviews, as they appear to have done), but otherwise I have no further comments.

- Thank you. We should have made it clearer in our previous response letter that we had in fact already also added these reliability estimates to the main text, in addition to reporting them in the letter.

Reviewer #1 (Remarks to the Author):

The authors have added new data, analyses, and discuss additional background articles. I very much appreciate those additions, as well as the authors' patient explanation of their approach! While I suspect that the authors might consider some of these additions unnecessary, but I really think that they will help making this work more digestible (also in its details) to a wider readership. I am convinced that this work will make a substantial contribution to the literature. I want to congratulate the authors to their nice work, and want to endorse it fully for publication!